# Magnetic voluntary head-fixation in transgenic rats enables lifespan imaging of hippocampal neurons

P. Dylan Rich [1] ✉, Stephan Yves Thiberge [2], Benjamin B. Scott [3,4,5], Caiying Guo [6,7], D. Gowanlock R. Tervo [6,7], Carlos D. Brody [1,8], Alla Y. Karpova [6,7], Nathaniel D. Daw [1,9] & David W. Tank [1,2,10] ✉

The precise neural mechanisms within the brain that contribute to the remarkable lifetime persistence of memory are not fully understood. Two-photon calcium imaging allows the activity of individual cells to be followed across long periods, but conventional approaches require head-fixation, which limits the type of behavior that can be studied. We present a magnetic voluntary head-fixation system that provides stable optical access to the brain during complex behavior. Compared to previous systems that used mechanical restraint, there are no moving parts and animals can engage and disengage entirely at will. This system is failsafe, easy for animals to use and reliable enough to allow long-term experiments to be routinely performed. Animals completed hundreds of trials per session of an odor discrimination task that required 2–4 s fixations. Together with a reflectance fluorescence collection scheme that increases two-photon signal and a transgenic Thy1-GCaMP6f rat line, we are able to reliably image the cellular activity in the hippocampus during behavior over long periods (median 6 months), allowing us track the same neurons over a large fraction of animals' lives (up to 19 months).

The hippocampus is crucial for episodic memory[1], a hallmark of which is that events can be remembered over a whole lifetime[2]. The precise neural mechanisms within the hippocampus that contribute to this remarkable persistence remain unknown[3]. Monitoring the activity of neurons over the whole lifetime of animals during ethologically relevant behaviors is a crucial technical advance that is needed to fill this gap.

Head-fixed preparations are widely used in neuroscience research, providing optical access to the brain for cellular calcium imaging[4] and optogenetic stimulation[5] as well as the ability to present controlled stimuli to animals[6,7]. However, such preparations severely limit the complexity and naturalism of behavior that can be studied.

The study of spatial navigation at naturalistic scales[8] and in complex environments[9], as well as episodic[10] and social tasks[11] requires the full response repertoire of a freely-moving animal. Rats can be trained to voluntarily head-fix[12], offering the best of both worlds: stimulus control and brain access during head-fixation, and the naturalism of freely moving behavior otherwise. Previous systems in rats[12,13] and mice[14–17] have relied on a mechanical clamp to restrain a head-plate attached to the head of the animal. Head restraint is unavoidably stressful[18,19], and animals must be gradually habituated to it, for instance requiring progressive increases in the pressure of clamping pistons. The ability of animals to self-release is also critical[12,17], but the mechanical solutions implemented to date impose a delay and are a potential point of

[1]Princeton Neuroscience Institute, Princeton University, Princeton, NJ, USA. [2]Bezos Center for Neural Circuit Dynamics, Princeton University, Princeton, NJ, USA. [3]Department of Psychological and Brain Sciences, Boston University, Boston, MA, USA. [4]Center for Systems Neuroscience, Boston University, Boston, MA, USA. [5]Neurophotonics Center, Boston University, Boston, MA, USA. [6]Janelia Research Campus, Ashburn, VA, USA. [7]Howard Hughes Medical Institute, Ashburn, VA, USA. [8]Howard Hughes Medical Institute, Princeton University, Princeton, NJ, USA. [9]Department of Psychology, Princeton University, Princeton, NJ, USA. [10]Department of Molecular Biology, Princeton University, Princeton, NJ, USA. ✉e-mail: dylan@dylanrich.org; dwtank@princeton.edu

failure; even one unsuccessful release attempt is enough for the system to become aversive for animals[17].

We present a new approach that improves upon existing systems by eliminating any actual restraint of the animal. Instead, animals are trained to clip into a magnetic coupling from which they can leave at any point. Instead of being trained to tolerate restraint, animals just learn to hold still. This system is fundamentally less aversive and, since it employs no moving parts, eliminates the risk that failure of the system could derail long-term longitudinal studies.

Cellular resolution two-photon calcium imaging requires high stability[4], and, for voluntary head fixation, micron-scale repeatable registration between insertions. Kinematic clamps are widely used in mechanical engineering to repeatably reposition objects in space[20], and when integrated into a voluntary head-fixation system allow in vivo two-photon calcium imaging[12,13]. We present a magnetic, full kinematic system with a geometry optimized for long-term voluntary use in rats. Ultra-hard and wear-resistant bearing surfaces provide the stability and reliability for routine two-photon imaging over long periods.

Due to the inherent optical sectioning of two-photon microscopy, every emitted photon from a sample is effectively signal[21]. Conventional two-photon microscopes use a single objective to both focus excitation light and collect emitted light; any emitted fluorescence that does not enter the front element of the objective with an appropriate angle is lost. Capturing these lost photons would improve the signal-to-noise ratio, increasing possible imaging depth and reducing the excitation power required (to prevent phototoxicity). Hybrid imaging/non-imaging objectives[22], fiber light guides[23], and parabolic epifluorescence reflectors[24,25] have all been used to successfully improve collection efficiency, however, existing designs require the addition of complex collection optics to the microscope. We present a simple epifluorescence collection scheme that is based on a parabolic reflector placed close to the sample. A single drop-in piece replaces the cylindrical cannula widely used for deep brain imaging[26] and redirects otherwise lost photons into the existing collection path.

Genetically expressed calcium indicators in transgenic animals are the preferred method for neuronal population imaging in mice[27,28], allowing large populations of neurons to be monitored over long periods[29]. Rats offer the opportunity to study more complex cognitive behaviors compared to mice but previous calcium imaging in rats has relied on the viral expression systems[12,13,30] or bolus loading[31] which limits the scope of experiments that can be performed. Transgenic rats have been developed[32] that express GCaMP6f under the Thy-1 promotor and demonstrated for cortical imaging. We report a line generated using this technique that shows strong, stable, and sparse expression in the CA1 region of the hippocampus, making it ideal for long-term population imaging of the same neurons.

We show that the magnetic head-fixation system achieves the stability and trial-to-trial reproducibility required for two-photon imaging; it is fast for animals to learn and animals are comfortable performing hundreds of trials over multiple sessions for months and years. Combined with stable transgenic expression of GCaMP6f in rats and the improved collection efficiencies of the epifluorescent collection cannula, we have been able to record from the same neurons in the hippocampus, during behavior, across the majority of animals' lifetimes, opening up a domain of experiments with profound implications for the study of brain function.

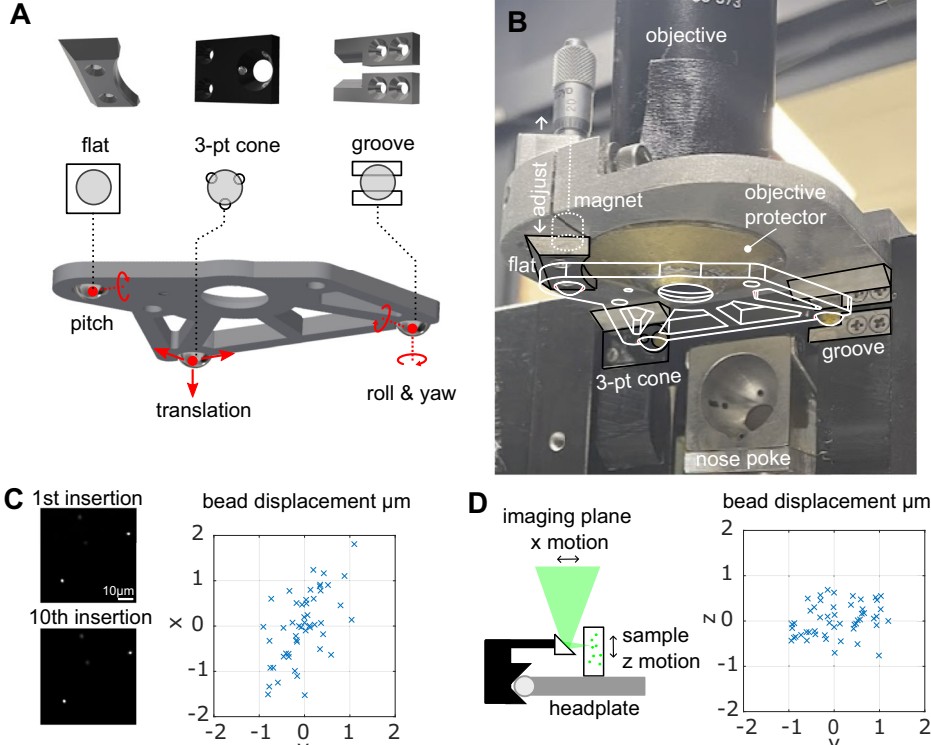

**Fig. 1 | Magnetic voluntary head-fixation system. A** The core of the system is the head plate with three ball bearings and three bearings of the Kelvin kinematic coupling. The 3-point cone, groove, and flat bearings non-redundantly constrain the six degrees of freedom of the head plate. **B** Photograph of the bearing system showing interface with head plate (white, wireframe). The brass objective protector prevents the animal from touching the front aperture of the objective while not occluding the optical excitation or emission. The magnet and adjuster are shown for the flat bearing only, two other magnets and adjustment screws sit behind the cone and groove bearing. **C** First and tenth images of a fluorescent bead sample taken during in vitro testing of kinematic registration and the bead displacements in x and y for all insertions (n = 50) during the test. **D** A microprism attached to the bearing system allows the translation of sample z-motion to imaging plane x-motion. The bead displacements in $x$ and $z$ are shown for all insertions (n = 50) of the test.

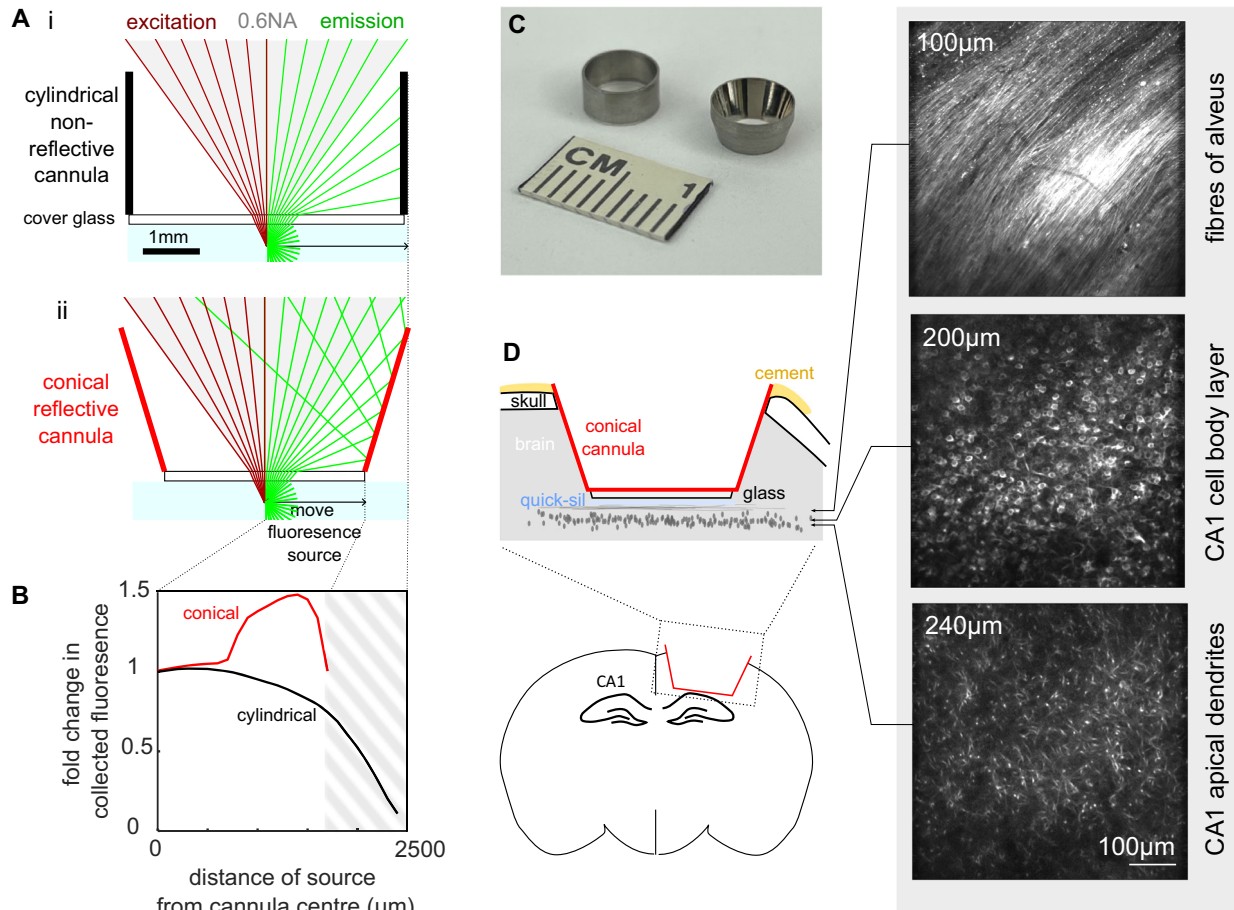

**Fig. 2 | Conical cannula increases collected fluorescence and allows imaging of GCaMP6f expression in dorsal CA1. A** Ray tracing diagrams for the conventional cylindrical non-reflective cannula (**i**) and the conical reflective cannula (**ii**). In both panels the fluorescent source is illustrated at the center of the cannula, below the cover glass, in water. On the left-hand side of the diagram the two-photon excitation light rays are shown for the 0.6NA air objective. On the right-hand side, the fluorescent emission rays are shown. Only rays that escape the glass-air interface are extended. Compared to a standard cylindrical cannula, the conical reflective cannula redirects emitted fluorescence that would otherwise be lost towards the front aperture of the objective, where it may be recorded (providing they do not exceed the spatial-angular acceptance of the collection optics). **B** Change in measured fluorescence at 150 μm below the cover glass (bottom of cannula, this corresponds to the typical depth of the CA1 pyramidal layer). The conical cannula provides up to 1.5× increased fluorescence collection compared to a standard conical cannula. The greatest improvement was measured towards the edge of the cannula. The fall off of the fluorescence for the cylindrical cannula is due to partial obscuration of excitation beam, which is also a factor for the conical cannula since they have a common top diameter. The hatched area indicates an invalid region of measurement for the conical cannula since it has a smaller bottom diameter than the cylindrical cannula. **C** Photograph of the cannulas. The inner surface of the conical cannula is polished to a mirror finish to improve collection efficiency. **D** Installation of the conical cannula over dorsal CA1. Schematic, *left*; two-photon images from an anaesthetized animal showing expression of GCaMP6f in the pyramidal cell layer at different depths below the bottom of the cover glass, *right*. Expression was similar to other animals (*n* = 7).

## Results

### Design of the magnetic head-fixation system

The design of the magnetic head-fixation system was based on two constraints, micron-scale registration of the head plate between insertions and ease of use for the animal. Micron-scale registration was achieved by kinematic design based on the Kelvin coupling[20]: an arrangement of a cone with three contact points, a vee-groove with two contact points, and a flat piece with a single contact point. These six contact points will non-redundantly constrain the six degrees of freedom of the head-plate according to the principle of exact constraint (Fig. 1A, B). We used hardened stainless steel or tungsten carbide for the bearing surfaces. Extreme hardness, wear resistance, and electrical conductivity make tungsten carbide the ideal material for this application, especially in the front bearings which are more subject to wear.

Ease of use for the animal means that animals should be able to easily insert and exit at will from the system. The nose poke (Fig. 1B) performed the initial coarse alignment of the head through the alignment of the snout. Forward translation of the head, achieved by moving the nose poke back, is sufficient to seat the two front bearings which constrain the translation, roll, and yaw of animal's head. The final pitch degree of freedom is constrained by the flat bearing; adjusting the pitch of the nose poke during training allows the animal to readily learn the correct position.

We used permanent magnets behind each bearing to provide a light attractive force on the bearing balls of the head-plate, both to aid the seating of the bearings and to help the animal keep the head-plate correctly seated during fixation. The system operates with no moving parts and animals could easily overcome the magnetic attraction and so break fixation at will. The magnetic force was started at zero initially and was increased by changing the distance from the magnet to the bearing ball with a micrometer screw. Animals were incentivized to complete fixations by delivering a chocolate milk reward at the end of the fixation duration.

### Registration is accurate in vitro

We assessed the reproducibility of the system without an animal by measuring the registration accuracy using a fluorescence bead sample

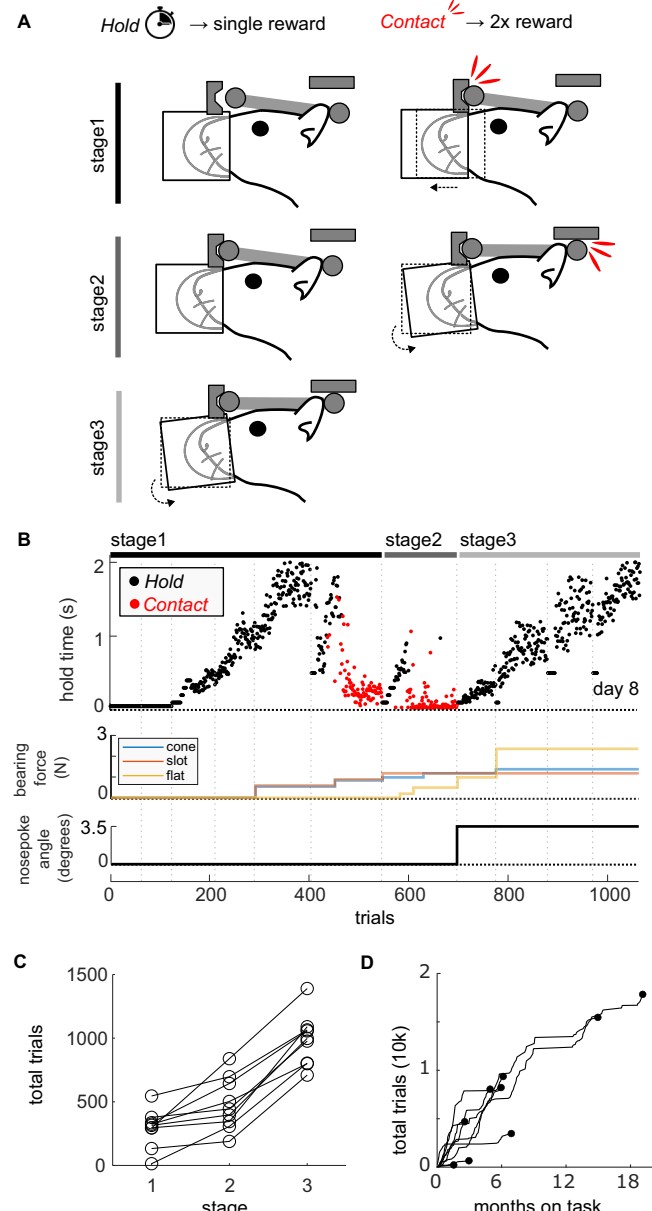

**Fig. 3 | Behavioral training of voluntary magnetic head fixation. A** Three stages of training. Animals can receive reward either by completing a *hold* for a specified duration or making a *contact* which leads to an immediate double reward. In the first stage, animals receive a single reward for staying in the nose poke for the hold time; by increasing the required hold time, and moving the nose poke, the animal makes contact with the front bearings, which leads to an immediate double reward. Once animals are reliably making contact, they move to stage 2 which requires a hold on the front bearings; by increasing the required hold time, and rotating the nose poke, the animal makes contact with the rear bearing, which leads to an immediate double reward. Once animals reliably make rear contacts, they move to stage 3 which requires a hold on all bearings; hold duration is increased gradually. The nose poke angle can be adjusted to help animals maintain the full hold. **B** Training profile for a single animal. First axis shows the hold time, each point is a completed trial, either for the experimenter-determined required hold time (black) or by making a contact which leads to an immediate reward (red). Second axis is the magnetic bearing force that was adjusted manually (typically at the start of a session). Third axis is the nose poke angle, which was adjusted only once to improve the hold duration at the start of stage 3. Vertical lines are separate sessions. **C** The total number of trials needed to complete each stage of training for each animal (*n* = 9). **D** The cumulative number of trials completed over time. Each line represents one animal (*n* = 9), with the dot showing the final total of the number of trials over months. Once achieving holds on all bearings of at least 2 s, animals are considered on task and are able to continue performing indefinitely.

conventional cylindrical cannula and found up to a 1.5-fold increase in the collected fluorescence (Fig. 2D). Painting the inner walls of the conical cannula matte black confirmed this increase was due to the reflective walls (Fig. S2A) and polishing the interior walls of the cylindrical cannula provided only a minimal improvement, confirming the contribution of the conical shape (Fig. S2B). We also saw improvements at different imaging depths (Fig. S2C). We saw the most improvement towards the edge of the cannula, with little improvement in the center (Fig. 2B). Ray-tracing simulations revealed that this was a consequence of the limited spatial-angular acceptance of the objective and collection optics (Fig. S3A)[34]. Reflected photons from the center of the cannula enter the front aperture of the objective with an angle and position that means they do not make it to the detector (Fig. S3C). The improvement was also evident in a scattering tissue phantom (Fig. S2A); the extra collection efficiently enables imaging of the CA1 pyramidal layer through the strongly scattering white matter of the alveus (Fig. 2E) which is thicker in rats compared to mice.

**Transgenic rats**
Expression of genetically encoded calcium sensors has previously been achieved in rats using injection of viral vectors such as adeno-associated viruses[12,13]. However, viral expression is difficult to maintain at healthy levels for long periods, severely limiting the scope of such experiments.

We used transgenic animals generated as previously described[32], which carry the gene for GCaMP6f under control of the Thy-1 enhancer. Different rat lines have different levels of GCaMP6f in different brain regions[28]. We selected a line, Thy1-GCaMP6f Line 8, that showed strong expression in the dorsal hippocampus (Figs. 2E and S4A). GCaMP6f was expressed in a sparse subset of hippocampal pyramidal cells in CA1 (Figs. 2E and S4B), and in vivo assessment of expression showed clear somatic localization (Fig. 2E).

**Animals are quick to learn magnetic head-fixation**
Naive animals were trained to successfully align and hold their heads in the kinematic mount for at least 2 s in 988, 203 (mean, sd) trials over 12.2, 4.1 (mean, sd) sessions. Animals progressed rapidly through the three stages of training (Fig. 3), quickly learning to hold a nose poke and make contact with the front bearings (stage 1); hold contact with the front bearings and make contact at the rear bearing (stage 2); and finally hold contact with all bearings (stage 3). The magnetic attractive

(Fig. 1C). Across 50 insertions we measured the RMS of the displacement to be 0.81, 0.47, and 0.36 μm in the *x*, *y*, and *z* dimensions, respectively. This is in line with previous kinematic registration errors using ruby ball bearings and steel stage bearings[33].

**Reflective conical cannula for enhanced epifluorescence collection**
Image quality in two-photon microscopy is improved with increased collection of emitted fluorescence. Previous attempts to capture additional fluorescence in two-photon microscopy have utilized reflectors and relays to redirect light to collection optics[22,25]. In this system, we modify the widely-used cannula prep for imaging deep brain structures such as the hippocampus[26] to optimize light capture. By using a conical (Fig. S1) rather than cylindrical cannula, and polishing the interior wall to a mirror finish, we formed a collection optic that approximates the ideal parabola[24] with a focal point at the imaging plane. Emitted light that would be otherwise lost is redirected up into the front element of the objective (Fig. 2A) and recorded with the existing collection optics. We measured the improvement in the optical collection of the polished conical cannula over the

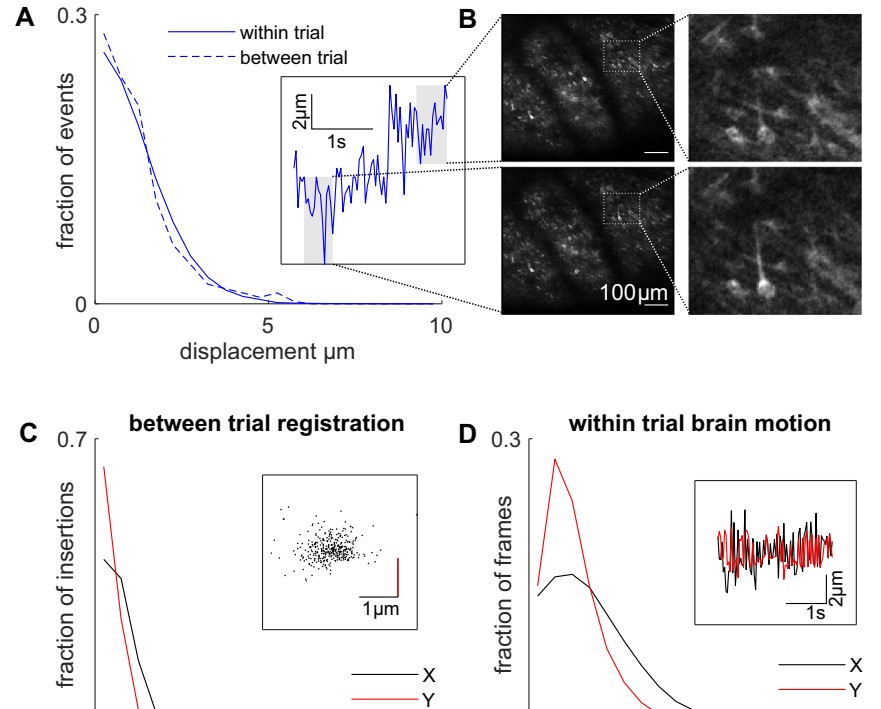

**Fig. 4 | Trial-to-trial registration in vivo. A** Histogram of the displacement of the brain in the *z*-axis, axial to the focal plane, for two sessions in two rats. Inset shows the calculated frame-by-frame *z*-axis displacement for a single trial with a large range displacement. **B** Shows the average fluorescence fields of view for two 0.5 s periods at either end of the *z*-axis displacement range for the single trial inset in A. At the extreme range of such displacements, the same fine structures such as

dendrites are visible in the image. **C** Histogram of the between trial registration displacements for six rats. Inset shows the spread of trial-to-trial displacements for a single session. **D** Histogram of the within-trial brain motion in *x* (anterior–posterior) and *y* (medio–lateral) for all rats. Inset shows the time course of the *x* and *y* displacements for a single trial.

force for each bearing was introduced after animals began to make contact with each bearing, and was increased incrementally over a few sessions (Fig. 3B), to a level that made it easy for animals to sustain fixation whilst still being able to break free (1.5–2.5 N per bearing; overall measured head-plate pull-out force <4 N). The gradual introduction of the magnetic force was implemented so that animals were not surprised by it, and we saw no evidence that it was aversive (animals continued to make fixation attempts). This magnetic force both helped the alignment into the bearings and aided retainment as small forces generated by the animal did not cause it to lose fixation. Once proficient, animals typically performed daily sessions of 100 s of fixations over multiple months and years, indicating a high degree of comfort with the apparatus (Fig. 3D).

**Fixation duration and variability**
We targeted a fixation duration of 2–2.5 s to allow for a 1 s odorant stimulus presentation and a short delay period, which was readily achieved in all animals. Indeed, animals remained clipped into the system after reward delivery and during consumption of the reward despite no contingency for doing so, further indicating that the fixation was not aversive (Fig. S5B). Since the animals were not restrained, it was possible for animals to break fixation before the target duration had been achieved, and they had to reinitiate another fixation attempt after a 1 s delay. Animals were able to complete the fixation on their first attempt in 76 ± 9% (mean over animals ± sd) of trials (Fig. S5).

By increasing the duration of the hold period before the reward was delivered, we were easily able to increase the fixations from 2.5 to 4 s for two animals within a single session (Fig. S5A, B). In a separate

experiment, we asked how long an animal would be able to remain head-fixed. We provided one animal with random intermittent rewards for as long as it remained head-fixed; it was able to remain head-fixed for up to one minute at a time (Fig. S6), demonstrating the level of comfort animals had with the system. This duration is comparable with previous reports in mice using a mechanical clamp and release switch[17] which allows multiple self-contained trials to be presented per fixation insertion.

**Trial-to-trial registration and brain motion in vivo**
Brain motion is a phenomenon of all in vivo imaging preparations and must be sufficiently small to be corrected by software. Image displacement, within trial (Fig. 4A, D, 1.64, 1.19, 1.67 μm RMS in *x*, *y*, *z* axes, respectively) and between trials (Fig. 4A, B, which includes any kinematic registration error; 0.55, 0.38, 1.50 μm RMS) was similar to reported values for awake behaving rats[12] and mice[4]. Motion within the imaging plane (*x*, *y* axes) could be corrected by existing non-linear motion correction algorithms (see Materials and Methods) leading to stable images (Svid_1). The influence of axial movement (*z* axes) cannot be corrected in software since the brain moves tangential to the imaging plane, however, the range of displacement is modest compared to the axial point spread function of the microscope and the diameter of cell bodies. Close inspection of images did not reveal obvious differences in the view of fine neuronal processes which are most sensitive to axial motion (Fig. 4B, Svid_1).

**Calcium transients and odor responses in CA1**
Over our large imaging fields of view, we were able to visualize the pyramidal cell body layer of CA1. As expected from the in vitro

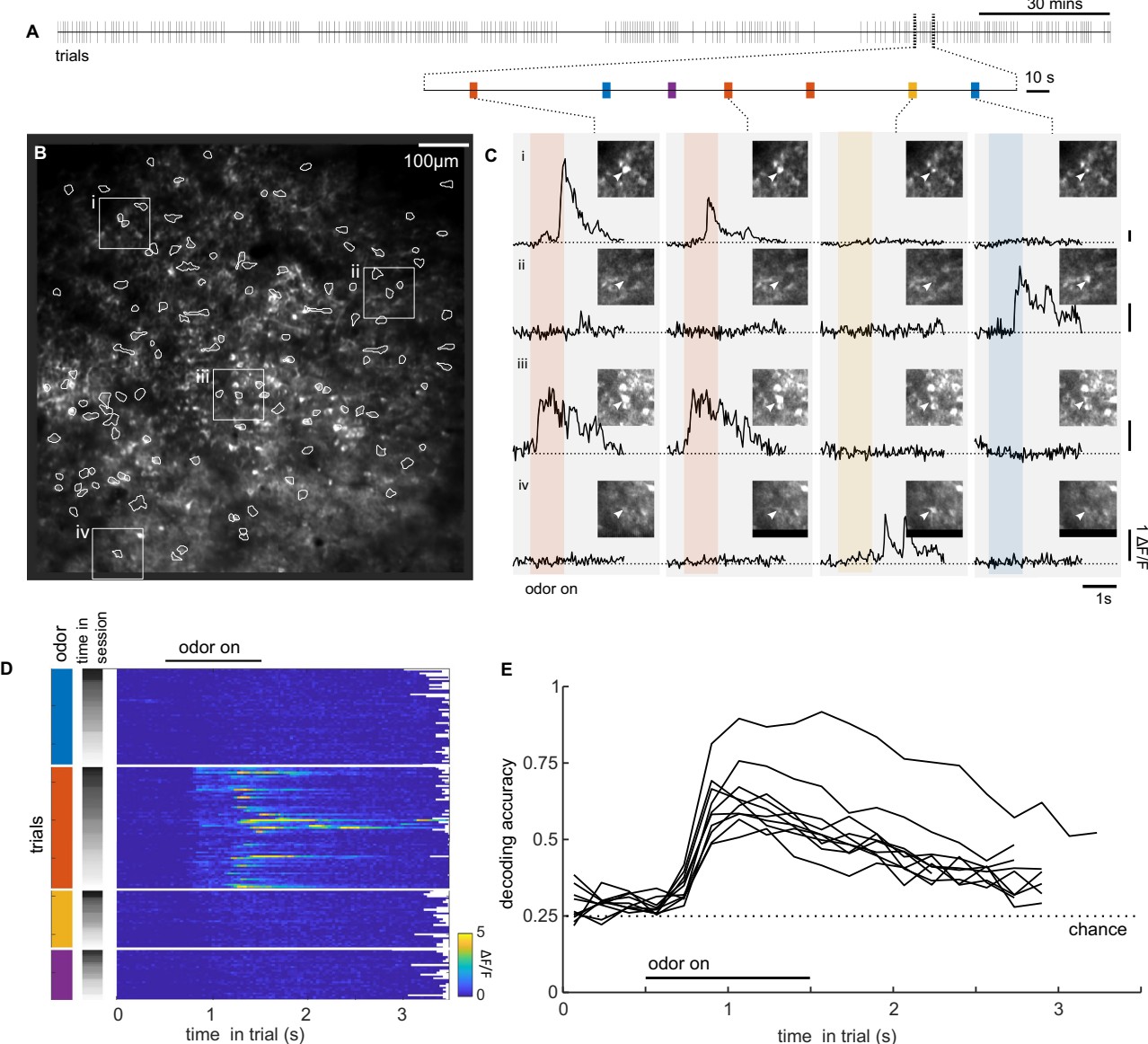

**Fig. 5 | Odor responsive cells. A** Timeline of a behavioral session, with tick marks showing individual trials. **B** Average fluorescence field of view during one experiment (out of 10 sessions). Active cells that were detected by the CNMF algorithm are circled. Numbered squared inserts (**i–iv**) correspond to regions centered on individual cells illustrated in (**C**). **C** Fluorescence traces of four cells (**i–iv**) over four trials with various odors presented. Colored bar shows the 1 s period of odor presentation; different colors correspond to different odors. **D** Fluorescence traces for one cell (**iii**) for all trials in the session. Trials are first sorted by which odor was presented, and next by time in the session. **E** Odor decoding accuracy for a cross-validated linear classifier over the population of cells for 10 sessions for four rats. Dashed line shows the chance decoding level expected by chance.

assessment, expression of GCaMP6f was sparse, with only a subset of cells being visible (Fig. 5B, Svid_2). We presented odorants to animals during fixation as part of a place-odor association task[35]: animals were trained that a large reward was available in a remote goal location depending on what odor was presented during fixation (Svid_2). We saw neurons in dorsal CA1 that responded selectively to different odors (Fig. 5A–D), and we were able to decode the presented odorant identity (Fig. 5E) at a population level.

### Long-term imaging
The expression of the calcium sensor and optical window clarity enabled imaging over a large fraction of animals' lifespans. We were able to image hippocampal activity during voluntary head-fixation up to 21 months following surgery when one animal was 25 months old (Fig. 6A). Cells continued to show clear calcium transients (Fig. 6D),

and the calcium indicator remained nuclearly excluded, indicating a level of GcaMP6f expression that does not affect cell function[27]. By returning to the same field of view, the same hippocampal cells could be tracked by comparing the morphology of individual neurons as well as the relative position of neighboring cells (Figs. 6B, C and S7E, F). We were able to track individual cells across a 19 months period in one animal (Fig. S7), other animals showed stability and longevity of the imaging window over many months (Fig. 6E, F).

### Discussion
Here we demonstrate three advancements that allowed us to track individual neurons over the entire adult life of animals as they perform a behavioral task: an entirely at-will magnetic head-fixation system that is safe and easy for animals to operate and is reliable over years of use, an epifluorescence collection cannula that improves two-photon

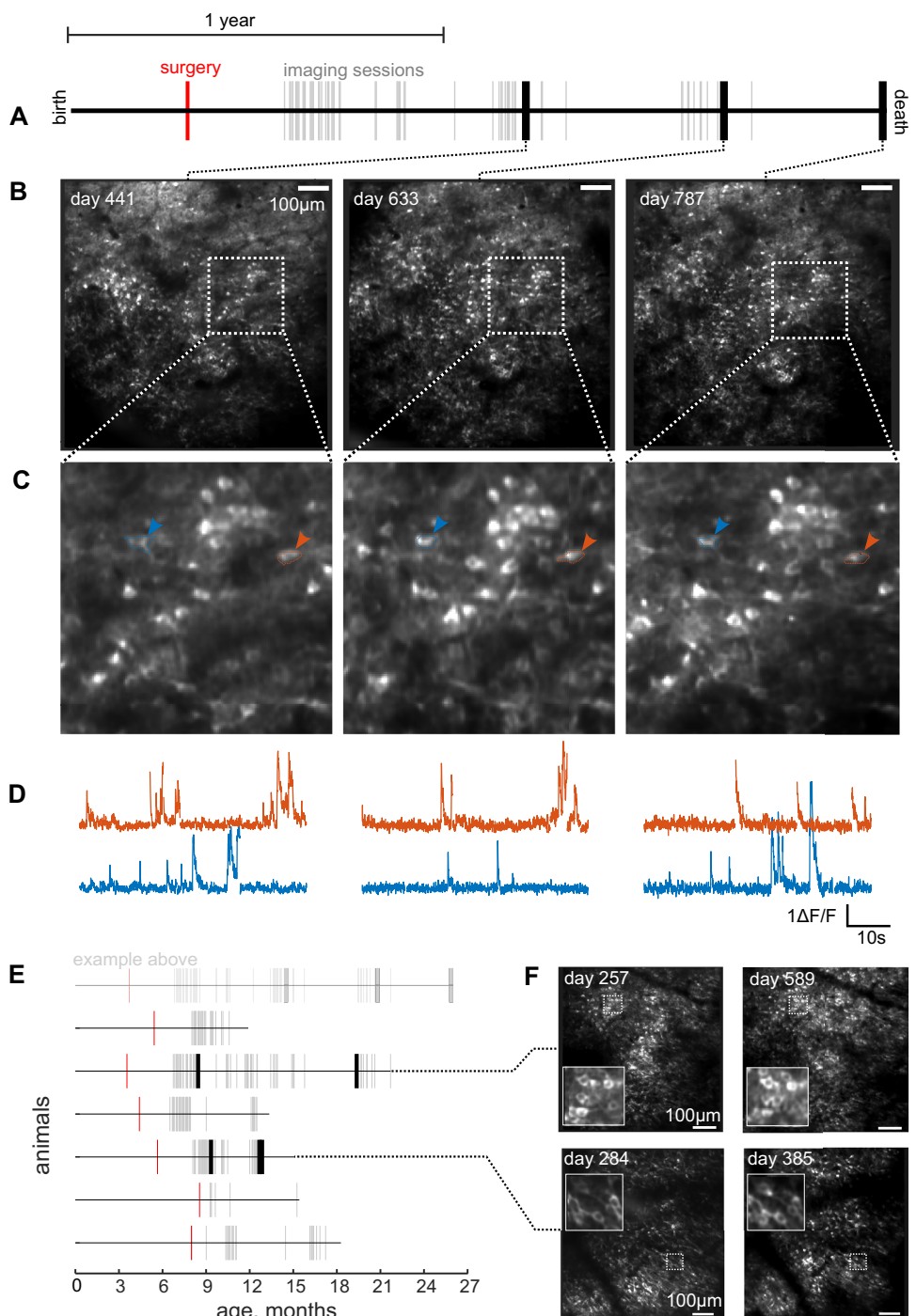

**Fig. 6 | Lifetime, longitudinal imaging of the same cells during voluntary head-fixation.** **A** Lifeline of one rat; line shows the extent of the animal's total life, 25 months. Awake-behaving imaging voluntary head-fixation sessions are shown as ticks. Bold ticks are example sessions illustrated below. **B** Average fluorescence fields of view for three example sessions. **C** Expanded views. Two corresponding cells are marked. **D** Calcium fluorescence traces for the two cells indicated above. Gaps in the calcium trace reflect the concatenation of individual trials. **E** Lifelines of other animals, as in (**A**). Each grey tick represents an imaging session where the animal was successfully head-fixing, performing the behavioral task, and calcium data were recorded, number of sessions for each animal = 60, 27, 62, 38, 44, 7, 23. **F** Fluorescence fields of view and zoomed insert for two sessions for two animals.

imaging signal, and a transgenic rat line that expresses GCaMP6f at stable levels in pyramidal cells of the hippocampus.

Involuntary head restraint is a widely used technique in neuroscience but has proved challenging to deploy with rats[18], leading to less widespread adoption compared to mice. The system described here is a substantial improvement over previous voluntary head-fixation designs that have depended on mechanical restraint of animals[12,14,15,17]. The system provides the registration accuracy required for two-photon imaging through a kinematic clamp design[12,33] using ultra-hard wear-resistance tungsten carbide and hardened stainless steel bearings. Since the system does not have any moving parts, it is also failsafe; there is no possibility of animals becoming trapped due to equipment failure. This allows the possibility of long-term, unsupervised, high throughput, home cage experiments with fewer safety

and welfare concerns than existing systems. Adapting the system for use in mice would require a reduction in scale and the magnetic forces, and is feasible given the previous demonstration of kinematic clamps[33] and voluntary head-restraint in this species separately[14,16,17]. Adapting this system for non-human primates offers a compelling direction for minimizing the invasiveness of these experiments, and could take advantage of the mature engineering principles required for designing high-load kinematics[20]. Although current head-mounted two-photon microscopes have begun to provide sufficient quality images to enable freely moving physiology[36], they cannot approach the flexibility and extendibility of a benchtop system. Sophisticated optical approaches such as mesoscale and multi-region imaging[37,38] two-photon optogenetic stimulation[39], and holography[40] and other exotic optical techniques for volumetric[41]and kilohertz imaging[42] and deep imaging using adaptive optics[43] can, at present, only be performed using benchtop systems and head-fixed animals.

The principle drawback to experiments using voluntary head fixation is that brain activity outside the period of fixation is not recorded. However, if a task is carefully designed, then the key decision-making period can be sequestrated within the head-fixation duration. In this, and previous work[13] the decision is reported by the animal by making a left or right turn immediately following fixation to a different nose pokes. In other work in mice, directional licking to two separate lick spouts has been used[17] so that the entire relevant task period is during fixation. Another drawback to our approach is the wearing of the kinematic components, which can become electrically unreliable despite satisfactory kinematic alignment. Ultra-hard tungsten carbide surfaces ameliorate this, providing longer service life; but independent optical or capacitive non-contact methods to sense alignment may be preferable. Repeated fixations may also introduce brain motion, but this was comparable to previously reported levels for involuntary head fixation.

Building on previous approaches to improve epifluorescence collection in two-photon microscopy[25] we were able to achieve a 1.5-fold increase in signal using an easy-to-fabricate cannula piece that approximates a parabolic reflector. We designed the current conical cannula shape for hippocampal imaging with a long working distance air objective but the current geometry would also be compatible with water immersion objectives commonly used for in vivo two-photon microscopy since they have a similar half angular aperture (37.0° for $NA_{water} = 0.8$, 36.9° $NA_{air} = 0.6$) and sufficient working distances. In the presented configuration, little improvement is seen in the center of the cannula due to the reflected photons having too large an angle to travel through the objective, collection optics, and to the PMT. By changing the cannula angle or relaxing the current manufacturing constraints of a straight wall and performing end-to-end optical simulations, signal collection and field flatness could be improved and customized for specific imaging locations. Since we also observed a modulation of the collection gain for imaging depths, optimization of cannula shape could also be performed to maximize collected fluorescence from different target depths

We add to the existing Thy-1 GCaMP6f rat lines previously reported[32] with a line that shows strong, sparse, and stable expression in pyramidal cells of the CA1 region. As well as being useful in our application of magnetic head-fixation, these current transgenic animals could be used to monitor calcium activity in freely moving rats using head-mounted two-photon microscopes[36,44], head-mounted single photon microscopes[30,45] or for fully head-fixed rat behavioral tasks[6].

Tracking activity in the same cells over long periods is challenging. In electrode recordings, gliosis reduces signal[46] and electrode movement changes the recorded waveforms of cells[47]. Reports of cells tracked over time have relied on stability of recorded waveforms[48], stable tuning responses[49], flexible electrodes[50], or algorithmic post-hoc correction of electrode movement[51]. However, these methods are all indirect and not robust enough to enable routine lifetime recording, especially in the presence of representational drift[52] or age-related changes in waveforms[50] reported.

Calcium imaging provides a direct view of cells. Single photon imaging has been used to track cells over multiple days[45,53] but is challenging over longer periods due to lack of optical sectioning, light scattering and its ability to only visualize active cells[54]. Using two-photon imaging, the morphologies of individual cells are clearly visible, which has enabled longitudinal structural imaging of dendritic spines in vivo for long periods[55,56]. Using genetically encoded calcium sensors the activity of the same neurons can recorded over months[57], and as we demonstrate here, for well over a year. A sufficiently sparse and stable expression of the calcium sensor makes an unambiguous determination of cell identity across sessions relatively straightforward based on the unique constellation of cell positions. Future efforts to track cells across extended periods will be facilitated by having a similarly sparse labeling strategy as we have presented here.

The ability to follow the same cells over the lifetime of the animal has profound implications for the study of aging, and particularly, the study of age-related changes in the neural representations of memory. For example, continual optical access to the hippocampus and other brain regions over the lifetime of an animal would allow researchers to link the cellular and behavioral dimensions of Alzheimer's disease to the in vivo measurement of amyloid plaques[58]. More generally, these techniques enables the long-term experiments that are necessary to link neural dynamics to the process of age-related cognitive decline[59].

## Methods

### Animals and surgery description

All procedures performed in this study were approved by the Institutional Animal Care and Use Committee at Princeton University (IACUC protocol number 1837) and were performed in line with the Guide for the Care and Use of Laboratory Animals (National Research Council, 2011). We used 9 (5 male and 4 female) rats that expressed GCaMP6f under the thy-1 promotor aged between 3 and 9 months at the time of surgery. Animals were generated as previously described[32], 7 animals were included in the long-term imaging experiment following initial training on head-fixation.

Rats underwent analogous surgical procedures as previously described in mice[26] to obtain optical access to the dorsal hippocampus. Surgery was performed under aseptic conditions, and animals were anesthetized with isoflurane (3.5% induction, 0.5–1% for maintenance). Animals received pre-operative buprenorphine (0.02 mg/kg) and either post-operative meloxicam (1 mg/kg) or buprenorphine (0.02 mg/kg) for analgesia. Body temperature was maintained with a heating pad (Harvard Apparatus). The skull was exposed and the periosteum was retracted. A craniotomy was performed on the right hemisphere over the dorsal CA1 region of the hippocampus (4.2AP, 3.0 ML) The overlying cortex was aspirated to exposure the fibers of the external capsule, which was carefully retracted to exposure the fibers of the alveus. A thin layer of Kwik-Sil (World Precision Instruments) was squeezed on the exposed area and the custom conical cannula (see below) was quickly lowered and cemented in place with adhesive cement (C&B Metabond, Parkell). A titanium base head-plate (2.3″ × 0.6″, 1/16″ thick, with 4–40 screw holes to later accept the kinematic head-plate) was mounted parallel to the stereotaxic frame and cemented to the skull. The cannula was held in place at its rim by three magnets (1/16″ diameter, D12-N52, K&J magnets) and was implanted centrally to the head-plate aperture with the cover-glass parallel to the stereotaxic frame. These steps resulted in the cover glass being parallel and concentric to the head plate for imaging. Following surgery animals were left for ~1 month before the imaging window was assessed under anesthesia. At this point, the kinematic head-plate (Fig. 1A) was installed into the base head plate. The kinematic head plate was custom designed with 0.125″ thick titanium using

finite element analysis (Inventor, Autodesk) to optimize rigidity to weight. Three hardened stainless steel (440c 1/4″ 9642K39, Mcmaster Carr) bearing balls were aligned to head plate and a thin bead of conductive paint was applied to the joint to ensure electrical continuity across the head plate. The ball-bearing balls were then attached to the head-plate using the same adhesive cement used in the surgery. The total weight of the head plates (base and kinematic) was 22 g.

## Conical cannula
The conical cannula was constructed from 420 series stainless steel (Mcmaster Carr) chosen for its combination of corrosion resistance, magnetic (for holding the cannula during implantation), and high surface finish achievable. The conical cannula had inner diameter at the top of 5.2 mm and an inner diameter at the bottom of 3.6 mm (full drawing Fig. S2). The conical cannula was turned on a precision lathe (10ee, Monarch), which allowed a high surface finish to be achieved on such a small piece. Following machining the interior of the cannula was polished with diamond polishing pins (3 mm EVE polishing pins, Rio Grande) and polishing paste (Simichrome, Happich). A 4 mm diameter circular glass coverslip (#1 thickness, CS-4R, Warner instruments r) was glued to the bottom with UV curing glue (Norland 81, Thorlabs). The cylindrical cannula used for testing had an inner diameter of 5.1 mm and a height of 2.6 mm and a wall thickness of 0.1 mm.

## Imaging system
We performed imaging with a custom two-photon microscope controlled with the Scan Image software (Vidrio) running in MATLAB (Mathworks). Laser illumination was provided by a Ti:Sapphire laser (Chameleon Vision II, Coherent) operating at 920 nm. We used a long working distance air immersion objective lens (20×/0.6NA/13 mm WD, Edmunds optics) and a GaAsP photomultiplier tubes (H10770PA-40, Hamamatsu). The beam power was modulated by a Pockels cell (350-80 LA BIC −02 Conoptics) and the power used for imaging, measured at the front of the objective, was 150–200 mW. Horizontal scans of the laser were achieved using a resonant galvanometer (Thorlabs). Typical fields of view measured ~600 × 600 μm and data were acquired at 30 Hz. The measured lateral and axial resolution of microscope given by the FWHM of the point spread function was 1.2 and 9.1 μm.

## Bead measurements
1 μm green fluorescent beads at a 1:1000 dilution were embedded in 3% agar gel for registration measurements. The agar solution was poured into a 5 mm diameter cannula with a glass coverslip glued to the bottom (Norland 91, Thorlabs) and allowed to set. For $x–y$ measurements the sample was attached to a head-plate which was inserted by the experimenter by hand into the bearing mount, 1 s (30 frames) of data was taken. The head-plate was removed and reinserted by hand between each acquisition. For measurement of z-registration error we attached a prism with a reflective hypotenuse to the bearing system (1.5-mm side length, BK7 glass, Optosigma), and oriented the bead sample 90° such that the glass coverslip bottom the cannula was oriented towards the prism[33], Movement in the z-dimension would be translated into movement in the x-dimension.

## Fluorescence collection tests
To test the collection gain of the conical cannula we used a dilute fluorescein sample (25 μM, Sigma-Aldrich) in a well with a cover slip glass over the top. Either the conical or cylindrical cannula, without a glass bottom, was placed on the top of the clover glass. We recorded 100 frames from the center of each cannula to its internal edge in 100 μm steps and did the same sequence at different depths below the bottom of the cover glass. The average raw fluorescence values at each depth and distance from the cannula center were normalized by the average fluorescence measured below the glass after removing the cannula and scaled to the value at the center of the cylindrical cannula

for comparison. For tests with investigating the effect of the reflective inner cannula wall, we painted the inner walls of cannulas with matte black paint. In this way we could directly compare the same cannula in a reflective state and a non-reflective matte state. For tests with a scattering sample, we created a tissue phantom with similar scattering properties as brain tissue[34]. We embedded 0.974 μm diameter polystyrene microspheres (Polysciences Inc., USA) at $5.4 \times 10^9$ beads/mL in 0.5% agarose and 10 μM fluorescein dye (Sigma-Aldrich). These parameters generate a tissue phantom that has a scattering length that is typical of mammalian brain tissue, ~75 μm at 520 nm[34].

## Ray tracing simulations
We conducted ray tracing in OpticStudio (Zemax). The available optical model of the 20 × 0.6NA objective from Edmunds Optics is a so-called black box and can only be used in sequential mode; a simulation of the full optical collection path was not possible in the required non-sequential mode. The key aspect of the collection optics and objective we sought to model was its spatio-angular acceptance, which we achieved with a pair of paraxial/ideal lenses and aperture stops as follows. The two-photon fluorescence excitation was a point source in water, with a glass coverslip followed by the cannula. A pair of paraxial/ideal lenses with a focal distance of 13 mm (WD of actual objective lens) were placed conjugate with the point source. At the first paraxial lens (objective) we placed an aperture stop with the same diameter as the front of the real objective (20 mm). At the conjugate image plane following the second paraxial (tube) lens, we placed a second aperture stop to model the combined spatio-angular acceptance of the objective and the collection optics. The total luminous flux was measured at a detector behind the image aperture stop as a function of the distance of the point source from the cannula center. We determined the diameter of the image aperture stop using the available black box model of the 20 × 0.6NA objective in a separate sequential ray-tracing simulation (Fig. S3A). In this simulation, the object size was increased until the chief ray exited the back aperture at 8°, which is the level of ~50% transmission of rays through the collection optics (based on the optical specifications of the microscope). This was assessed with the inclusion of a 200 m tube lens to form an image and using the *incident angle vs image height* function in OpticStudio (Fig. S3A, B). This object size represents the effective field of view from which light at the object plane will propagate to the PMT, and was determined to be 1.5 mm half-diameter. Since the configuration of our non-sequential objective and tube lens simulation had a 1× magnification, an aperture stop at the image plane would simulate the effective field of view.

## Magnetic at-will head fixation system
The kinematic magnetic head fixation system was designed to be structurally rigid and simple for animals to self-align. The retainment of the head plate within the bearings was facilitated with permanent magnets. The three bearings surfaces were made of hard, electrically-conductive materials. The three-point cone uses three tungsten carbide balls (0.094″, 9598K56, Mcmaster Carr) glued into an anodized aluminum bracket that arranged the balls in tetrahedral arrangement with respect the bearing ball in the head-plate; the groove bearing was made of hardened 440c stainless steel with some versions having a press-fit tungsten carbide insert (rectangular bar stock, 1690T11, Mcmaster Carr); the flat bearing was constructed of hardened 440c stainless steel. Behind each bearing was a neodymium magnet (D44-N52, K&J magnets) glued to micrometer screws (Mitotoya 148-202FT-H, Thorlabs) which were used to finely and reproducibly adjust the strength of retainment of each ball bearing. Following extended periods of use (many months, corresponding to tens of thousands of individual fixations), we noticed the electrical continuity detection could become unreliable despite reliable kinematic seating of the head plate (as assessed by imaging). Inspection of the bearing balls and stage bearings (Fig. S8) revealed this was likely due to wear on the

bearings causing some change in electrical properties of the surfaces. Replacement of the bearing components resolved the issue. The tungsten carbide bearing inserts on the groove bearing did not exhibit the same level of wear.

A spring scale (NK-20, M&A Instruments) was used to calibrate the retainment force of the micrometer settings. Contact with the each bearing and the bearing ball of the head plate was measured by a low voltage (15 v at 68 mA) electrical continuity circuit. The circuit detected which of the three bearings were seated corrected. If all three bearings were seated correctly, then a signal was sent to open the shutter on the imaging laser, begin image acquisition, and update the behavioral system. The position of the stainless-steel nose poke was controlled by mechanical positioning stages (front and back ET-50-x1, Newark systems; lateral TARA-4010, OptoSigma; vertical TAR-34403L, OptoSigma) and single-axis goniometer stage (GOH-40A51UU, Opto-Sigma), it allowed the coarse alignment of the animal's head and positioned a lick tube at the correct position so animals could lick while still head fixed. An infrared photodiode and phototransistor pair were used to sense the animal's snout in the nose poke. An objective protector was made of thin brass sheet bent and soldered into a truncated cone, and protected the objective from the animal, whilst allowing free optical access to the cranial window. To ensure a good electrical, mechanical, and magnetic interface, we cleaned the bearing balls on the head-plate of the animal and the bearing surfaces of kinematic mount before each session with a small amount of ethanol on a cotton applicator or optical tissue.

## Behavioral training

Animals were food restricted to 85% of their free feeding weight prior to behavioral training and maintained at this level during training. For some period when animals were not being trained, they were given access to food ad libitum and returned to food restriction prior to training. Animals weighed between 350 and 600 g throughout the experiment.

Stage 1—Animals were first trained to poke at the stainless-steel nose poke to receive a chocolate milk drop reward (1 drop, Ensure milk chocolate flavor nutritional shake, Abbott). Alternation with a nose poke at the rear of the chamber where they also received a chocolate milk drop. The length of time the animal had to hold its nose in the nose poke was slowing increased; the hold time was drawn from a distribution of times, the bounds of which were increased during the session. If the animal was successfully making holds, the lower bound was increased 0.1 s and the upper bound 0.2 s approximately every 10 trials; for a given session, this increase was stopped when the lower bound reached the value of start of session upper bound. This ensured that the animal did not have too large a jump in hold requirements throughout within a session. During the hold period an audible frequency modulated tone was played to indicate to the animal an ongoing successful hold. The nose poke position was adjusted backward so that the head plate bearing balls would contact the two front bearings which triggered a larger immediate reward (2 drops).

Stage 2—Contact with the front bearings now initiated the hold period which, as before was drawn from a range which was increased throughout a session. Contact with the rear, flat bearing now triggers the immediate large reward. In this condition the nose poke may be needed to adjusted in the pitch axes using a goniometer stage to enable contact with rear flat bearing. The holding force of the magnetic front bearings was increased in this stage, beginning at 0 N it was increased until the animal was able to maintain the hold (up to 2.5 N) and was increased in small increments every 10–20 trials over the course of 1–2 sessions.

Stage 3—Contact with all bearings now initiated the hold period which, as before was drawn from a range which was increased throughout a session. Animals were required to maintain fixation for the hold duration to receive a reward, a tone indicated that the animal

had maintained the required hold duration. The magnetic attraction for the rear bearing was adjusted at this stage, beginning at 0N it was increased until the animal was able to maintain the hold (up to 2.5 N) in small increments every 10–20 trials over the course of 1–2 sessions. The overall measured total head-plate pull-out force never exceeded 4 N for any of the animals, remaining below the level where animals would have potential difficulties disengaging from the bearing; rats can generate pull forces around double their body weight[60]. The lower bound of the hold duration was increased so that the hold period was fixed at a set value of 2–2.5 s. The angle of the goniometer could also be adjusted in this stage if the animal had difficulties maintaining the fixation. During task performance, we gave extra reward delivered with a small random exponential delay (exp mean 0.2–0.5 s) on a random 50% of trials while the animal maintained fixation. If animals did not complete a fixation (i.e., they did not remain fixated for the required hold time or make a contact to get a large reward during stages 1 and 2), there was a 1 s time-out period before a new fixation could be initiated. The illumination of a blue LED light inside the nose poke indicated to the animal when a new fixation could be initiated. Animals were given 10 "warm-up" trials at the start of each session, these were trials where the hold duration (typically 2 s) was shortened to 0.5 s.

## Odor guided navigation task

Once animals had learned to reliably clip in to the head fixation system they were trained to perform an odor guided navigation task. Animals were trained that different odors corresponded to different reward location in a maze constructed of enclosed linear track elements. We delivered odors (acetic acid 20%, 2-propanol, propyl acetate, anethole, trans-cinnamaldehyde, nutmeg, cardamom, clove, rosemary, star anise) to animals using a custom built olfactometer. Air was directed to flow though vials containing odorants using PTFE valves (Neptune Research) at a flow rate of 0.5 L/min. At the start of head fixation, blank air was delivered for 0.5 s, followed by 1 s of odorized air, followed by a scavenger vacuum through the system to evacuate any residual odorant for the next trial.

**Motion correction.** Imaging data were corrected for non-rigid brain motion using custom written MATLAB code based on a similar algorithm as NoRMCorre[61–63] in which the image is divided into overlapping patches and a rigid translation is estimated for each patch and frame by aligning against a template. The translations are then interpolated to create a motion field which is then applied to smaller overlapping patches. The maximum displacement out of all patches in a frame was used to calculate the displacement values for the $x$ and $y$ axes.

## Z-motion estimation

Motion perpendicular to the imaging plane (z-motion) was estimated by comparing imaging frames to a reference z-stack collected during anesthetized imaging session in two animals. The reference Z-stack was collected at the same coordinates as awake imaging and each image was made as the average of 100 frames taken at 1 μm spacing. In order to account for the non-ridged motion correction patches of the average motion-corrected movie were correlated with corresponding patches in the reference Z-stack. The Z-motion for each frame was calculated as average frame of peak correlation for each patch.

## Roi extraction

Fluorescence traces corresponding to individual cells were extracted from the motion corrected images using constrained non-negative matrix factorization algorithm (CNMF)[64]. Initialization of the spatial components for CNMF was done as previously published as was classification of identified components into cell-like and non-cell-like categories[61,65]. All components were manually inspected and reclassified if needed. $\Delta F/F$ for each cell was calculated using the modal value of fluorescence in 3-min long windows as baseline

fluorescence. Since CNMF only extracts cells that have calcium activity during imaging, silent cells that did not show activity during imaging were not included.

## Field of view registration

The first step that is required in order to return the same field of view is to have a calibrated zero position for the system as a whole. This was achieved with a plastic fluorescence slide that was permanently mounted to a dedicated kinematic head-plate. The fluorescence slide was a scored with a razor blade to generate unique and identifiable surface features. At the very start of all experiments, the artifact slide was inserted into the kinematic clamp and imaged. A zero position was chosen that had detailed surface features. Before each experiment, the artifact slide was reinserted, the microscope stage was moved in X, Y & Z to match a reference image of the zero position, and the coordinate system zeroed. This procedure positions the microscope to the same position, in the same coordinates, relative to the kinematic clamp. If the angle of the objective was altered (which was rare), it was adjusted such that the interface of the flat fluorescence slide appeared across the imaging plane at the same time as the microscope was stepped down through Z.

To match fields of views in the same animals across sessions, we could first return to the same coordinates of the target field of view (FOV; given the system registration defined above). Fine adjustment of the microscope position was then performed using the Scan image template matching feature. A reference Z-stack had to be taken during an anesthetized imaging session for a given FOV. The fine adjustment of X, Y and Z position was performed manually in the first ~10 trials of a session to maximize the cross correlation of the current FOV to a defined slice in the reference Z-stack.

To align fields of view within the same animal offline, we calculate the 2D cross correlation between the two average fields of view after motion correction. If imaging sessions were taken at different zoom levels, they were resized to a common scale using the MATLAB function *imresize*. In order to improve the robustness of this alignment, we subtracted the 2D cross correlation of Gaussian blurred images (sigma 4 px), this ensures that the peak in the resultant adjusted cross correlogram is determined more by high spatial frequency similarities which are indicated of aligned anatomical features. The peak in the resultant adjusted cross correlogram is taken as the alignment offset.

We identified active cells across imaging sessions by identifying candidates based on the overlap of the ROI mask generated by the independent ROI extraction. The criteria used was the if overlap area of the ROIs was greater than 50% of the area of each individual ROI. We then manually inspected pairs for similar morphology and position relative to other cells and features such as blood vessels.

## Odor classifier

We used a linear classifier approach to decode the odor identity. We used K-fold cross validation and trained a linear decoder on the population activity for cells that exceeded an average activity threshold of 0.03 $\Delta F/F$ across the whole session.

## Histology

Histology was performed to assess the raw GCamp6f signal[28]. Animals were anesthetized with an overdose of Ketamine Xylene and transcardially perfused with PBS and 4% Paraformaldehyde. Brains were extracted, and left in fixative for 12 h and then transferred to 30% sucrose in PBS until the brain had sunk. Brains were sectioned on a freezing microtome (Leica) in 50 µm sections and floated onto slides from a solution of DPBS +calcium (Gibco, ThermoFisher). Slices were mounted on slides with prolong diamond +DAPI (Invitrogen, ThermoFisher) and left refrigerated for at least 12 h before imaging. Slides were imaged with an epifluorescence microscope for whole brain

expression pattern and a confocal microscope for cellular resolution images.

## Statistics & reproducibility

No statistical method was used to predetermine sample size. Only neurons with at least one calcium transient are able to be identified by CNMF. Therefore, cells in the hippocampus that were silent for entire imaging sessions were not included. The experiments were not randomized. The Investigators were not blinded to allocation during experiments and outcome assessment.

## Reporting summary

Further information on research design is available in the Nature Portfolio Reporting Summary linked to this article.

## Data availability

Full schematics and design files for the voluntary magnetic head-fixation system are available at the following address. https://github.com/dylan2106/Magnetic-Voluntary-Head-fixation. The source data generated in this study have been deposited in the Zendo database https://doi.org/10.5281/zenodo.10651825, and are provided with this paper. Thy1-Gcamp6f- 8 rats are available from the Rat Resource & Research Center (www.rrrc.us submission #1010). Source data are provided with this paper.

## Code availability

Software for the voluntary magnetic head-fixation system are available at the following address. https://github.com/dylan2106/Magnetic-Voluntary-Head-fixation. Software for the ray tracing experiments is available at the following address. https://github.com/dylan2106/two_photon_collection_cannula.

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

## Acknowledgements

We thank A. Song for assistance with the microscope, J. Gauthier for help with hippocampal surgeries, and S. Koay for cell extraction software and assistance. We thank S. Lowe for assistance machining and H. Payne for comments on the manuscript. This work was supported by the Simons Collaboration on the Global Brain (D.W.T.).

## Author contributions

P.D.R. performed the experiments and data analysis, P.D.R., S.Y.T., and D.W.T. designed the experimental setup, B.B.S., C.G., D.G.T., C.D.B., A.Y.K. Generated and screened the transgenic animal line, P.D.R. wrote the manuscript with input from D.W.T, N.D.D., and D.W.T. supervised the project.

## Competing interests

The authors declare no competing interests.
