## [Peer Review File · Nature Communications]

REVIEWER COMMENTS

Reviewer #1 (Remarks to the Author):

The authors of this manuscript describe three strategies for 2P imaging, including a mechanism enabling voluntary head-fixation of rats for seconds to minutes during behavior tasks, a custom chronic window to increase fluorescence detection efficiency, and a transgenic rat line that allows lifespan calcium imaging in hippocampus CA1. While these improvements may be useful for specific applications, I find it challenging to discern the significance of these ideas for broader applications in the field, their innovation compared to current state-of-the-art methods, and the logical relationship between these three aspects—specifically, the necessity to combine them in a single paper. Given these concerns, I would not recommend publication, at least not in journal NC.

Major issues.

I. Voluntary head-fixation part

a. The authors describe a significant advantage of their method, emphasizing reduced stress and ease of use for animals. However, the absence of quantitative data throughout the paper to measure animal behavior during training and experiments makes it challenging to assess whether this method outperforms previous head-fixation techniques, including standard involuntary head-fixation or mechanically clamped voluntary fixation.

b. The authors claim that previous methods required gradual habituation, but Fig 3B-3C indicates that over 1,000 trials are needed for animals to assume the correct position, which is a substantial training burden. There is no clear evidence of marked improvement in convenience.

c. The quantitative analysis of variability in holding periods for animals is missing. It's crucial to determine the average holding time within individual animals and across multiple animals, along with its deviation. Inconsistent holding times could be problematic for experiments requiring precise timing, such as closed-loop photostimulation experiments and delay-decision tasks.

d. The repeatability of head-fixation appears convincing but relies heavily on maintaining the stability of the entire system (i.e., x, y, z, and angle positions of the objective and imaging FOV) throughout the experiment period, which spans 1-2 years. Achieving sub-micron level of stability is almost impossible, especially in shared benchtop systems. More importantly, authors did not explain how to use this method for multiple animals when the FOV, imaging depth, or imaging angle are completely different? Did authors mark a specific coordinate for each animals, and adjust the microscope before each experiment? If so, it is hardly to achieve so high repeatability without an online checking, which of course is not compatible with so short head-fixing period (1-2s).

e. Even the reliability of the head-fixation is convincing, but it seems highly depends on the standing-still of the entire system, i.e. the x,y,z and angle position of the objective and imaging FOV should not be moved during the entire experiment periods (1-2 years). I am afraid this condition is hardly achievable in most of the lab since the benchtop system is usually shared. Even more, how does the author do

multiple animals, if the FOV, imaging depth, or even optimized angle are different? One solution may be that a specific coordinate can be assigned to each animal, and before each experiment, the system needs to adjust accordingly. But in this way, the registration accuracy is then decided by the repeatability of the imaging system, not the holding mechanism, which will be way more worse. This issue needs to be addressed or at least discussed.

f. The authors mention head-mounted miniscopes for both 1P and 2P systems, which in my opinion could potentially address many of the issues raised. Therefore, the advantages of voluntary head-fixation over miniscope-based freely-moving animals need further demonstration, particularly as newer miniscope systems published in recent years with enhanced scalability and capabilities. Related to e. The biggest advantage of miniscope is that it can continuously recognize for the entire session in any paradigms. But this voluntary head-fixation seems can only be used in a very short fraction of the session, so it can only be used to study very specific questions. The limitations of behavioral paradigms are not discussed in the paper.

II. The conical cannula part

The key result supporting improved collection efficiency in this method is Fig 2D, which appears confusing. It's unclear why the improvement is only observed at the edge (>500um from the center). One possibility is that this improvement results purely from the cropping effect in conventional cylindrical cannulas, which would be naturally resolved by using a conical shape. The contribution of inter-wall reflection remains unclear, requiring more detailed information, such as a comprehensive Zemax/ray tracing analysis of the entire light pathway from the objective to the PMT. Furthermore, conducting simulations in scattering tissue to confirm that this improvement holds within the brain is advisable.

III. Lifespan imaging part.

The paper lacks quantification regarding how the same cells are defined, the fraction of cells that can be reliably registered in life-long recordings, and whether this ratio remains stable over different recording intervals. This information is essential for evaluating the reliability and utility of lifespan imaging. Additionally, it is unclear whether all the recordings presented in the paper were conducted under magnetic head-fixation or if these two aspects were separated.

IV. The logic between these three parts.

The paper suffers from a lack of logical cohesion between the three components: voluntary head-fixation, conical cannula, and lifespan imaging. It appears that these components were assembled without adequate consideration of how they complement each other. Therefore, maybe authors should consider to split them into three separate papers or show a more convincing biological application that cannot be conducted without one of them.

Minor:

In abstract "Existing techniques to record neurons in animals are either unsuitable for 17 longitudinal recording from the same cells or make it difficult for animals to express their full naturalistic behavioral repertoire." the authors should highlight other advantages of their solution instead of claiming that existing techniques make it difficult for animals to express their full naturalistic behavioral repertoire, given the existence of miniscopes.

Line 43-45 "but the mechanical solutions implemented to date impose a delay and are a potential point of failure; even one unsuccessful release attempt is enough for the system to become aversive for animals". It's essential to address how their solution avoids these potential pitfalls, especially considering variations in rat size, behavior, and power differences. I can imagine authors still need a lot of trial-and-error and need to adjust the magnetic force rat by rat. During these pilot trails, animals may be stressed.

Line 71-73 "Rats offer the opportunity to study more complex cognitive behaviors compared to mice but previous calcium imaging in rats has relied on the viral expression systems". This may not be entirely accurate. See these papers from Kerr lab (PNAS 2014 and Nature Methods 2020).

Line 97-98 "evidenced by intermittent failure of the electrical 98 contact mechanism despite sufficient kinematic alignment". I am not sure if I understand what authors described here. Including a figure and a dedicated method section would enhance clarity.

Line 179 "The gradual introduction of the magnetic force meant that animals were not surprised by it, and we saw no evidence 180 that it was aversive. " Authors should show supporting data and quantitative analysis are required to validate this assertion..

Fig.2B. This figure appears to be an illustrative cartoon, and it does not seem to represent an actual ray-tracing simulation. Given that the authors mention the use of an air objective, the angle of ray traces shown here seems already beyond the total-internal reflection criteria and should not be able to exceed the glass window. To ensure clarity and accuracy, it is recommended that the authors provide a genuine ray-tracing simulation that reflects the conditions mentioned in the manuscript. This would help readers better understand the optics involved in their methodology.

Reviewer #2 (Remarks to the Author):

The manuscript "Magnetic voluntary head-fixation in transgenic rats enables lifetime imaging of hippocampal neurons" by P.D. Rich and the coauthors is well written and clearly organized. In this manuscript, Rich et al. introduce a novel approach called the "magnetic voluntary head-fixation system" for conducting voluntary head-fixed, long-term two-photon imaging in transgenic rats. The paper presents several significant technical advancements compared to existing methodologies: 1) Kinematic Mount with Kelvin Coupling: The authors employ a kinematic mount based on the Kelvin coupling, providing micron-scale registration accuracy between insertions. 2) Magnetic Head Fixation: Instead of using conventional mechanical clamps, the authors utilize a magnetic fixation approach. This magnetic system offers several advantages, including ease of use for animals, reduced aversiveness, and the elimination of potential failures associated with moving parts. 3) Reflective Conical Cannula: To enhance

emission signal collection, the authors introduce a reflective conical cannula. This modification significantly improves signal collection efficiency compared to traditional cylindrical cannulas.

The authors apply their magnetic head-fixation system to Thy1-GCaMP6f transgenic rats and demonstrate the system's adaptability and long-term stability over the animals' lifetimes. Two-photon imaging is conducted during fixation while the rats perform a place-odor association task, leading to the identification of odor-responsive cells. This work holds substantial promise for the field of neuroscience, making it well-suited for publication in Nature Communications. However, there are several important issues that need to be addressed to enhance the manuscript's quality.

- 1) Figure 2B: It is advisable to label the schematics in Figure 2B as "Nonreflective Cylindrical" and "Reflective Conical" to clarify the critical role of the reflective inner surface in enhancing signal collection. Additionally, it would be valuable to assess how much the conical shape improves signal collection compared to both cylindrical and conical shapes with reflective inner surfaces. Performing non-sequential optical simulations and comparing them with experimental data could provide valuable insights.
- 2) Figure 2D: Provide information on the internal diameter of the cylindrical cannula and how it compares with the conical cannula in this test. Explain why the red plot ends before reaching 2 mm while the cylindrical one reaches almost 2.5 mm. Address whether the excitation beam cone is obstructed by either of the two cannulas during this test. Also, ensure consistent terminology by using "Nonreflective Cylindrical" and "Reflective Conical" instead of "Cylindrical" and "Conical."
- 3) Figure 2D's Caption: Clarify whether the signal is measured at 150 μm below the cover glass as stated in the caption or if the average signal from different depths is quantified, as mentioned in the Methods section. Discuss whether the improvement in collection efficiency is sensitive to the imaging depth and, if so, identify the optimal depth.
- 4) Figure 3B: Address the observed transition between stages 1 and 2, which appears abrupt and suggests a possible nonvoluntary factor such as applied force. Provide data on the force applied during the training stages and its duration, accompanied by the original Figure 3B for clarity.
- 5) Headplate Weight: Specify the weight of the headplate used in the experiments.
- 6) Magnetic Force and Contamination: Discuss the potential issue of small objects, such as bedding debris, contaminating the bearing surfaces and possibly interfering with the magnetic connection. Clarify whether such occurrences happened during the experiments and their frequency.
- 7) Baseplate and Cannula Alignment: Explain the methods employed to ensure concentricity and parallel alignment between the baseplate and the cannula during the craniotomy process.

Reviewer #3 (Remarks to the Author):

This manuscript describes a system for automatic head fixation of rats that employs permanent magnets. the system is an adaptation of a previous publication from the group on a kinematic mount for headfixing mice that was published in the journal of neuroscience methods by Scott. This paper by the

originators of the automatic headfixation technique in rats significantly extends the mouse magnet work because now they show the system is viable for collecting long term gcamp fluorescence data. Light collection is improved using a new canula.

This is exceptional work and represents a direction the field should be going-automated experiments controlled by the animal. In general is put together well and contain strong data showing feasibility. However, I feel as a method's paper it falls short because there is not enough information here for one to replicate the findings. There are some images of the system, but full mechanical drawings, software, and procedures are lacking they should be made available and linked to the submission directly.

It should be clearly stated that the magnets employed are permanent and not electromagnets, minor.

Registration and motion correction is brought up, but it is not clear how registration is done for repeated 2-photon imaging, and if it would be possible to automatically register animals that are imaged on different days? the paper implies that this is done, but there should be details given as this is key, also please share software for registration and control (is this via scanimage)?

Figure 3 is interesting, showing hold time. Data in panel B and total trials over up to 18 months, which is absolutely unprecedented. However, there is no mention of the sample size for these measurements (animal number for B,C, and D please). Is this a single rat or is it group data.

the trials are also relatively short in duration only 2 seconds. I think we would like to see a distribution of trial length across the different animals, what duration do rats prefer? It would also be good to separate male and female data.

There is a section on long fixation. But this doesn't have any sample sizes, so we don't know if this is one rat or a group, this should be clearly indicated and all data shown.

Fig S2 is nice for long holds but this is just one rat and it does not indicate a distribution of fixation times, based on the data shown 37 sec was the longest hold time. Please show more stats. Very short hold times may limit the utility of the method as movement related activity will likely contaminate recordings. please discuss and present all long hold data. Ideally, one would expect to see a table with all of the animals and something like the duration of fixation, numbers of trial, etc like the data in Fig. 3 but group stats.

The method section needs some addition to work to better. Describe novel elements and critical components. Many items are mentioned by supplier only, no catalog number or model number, the J Neurosci Meth article is better done in this regard. This includes the permanent magnet cat # and micrometers as well as critical ball bearings.

The possibility of using similar mounts on mice is also mentioned but I feel given the work that has been done by some of the co authors in mice that the issue of mice could be addressed more thoroughly, how would the rat system shown be modified and what are the holding forces etc? Can mice be trained for long holds?

Provide a bit of information about the air objective used X,Y, Z resolution as they have used it for 2P in the current rig.

Minor

“Figure 6 - Lifetime imaging of the same cells” lifetime imaging could mean fluorescence lifetime so this might confuse some?

REVIEWER COMMENTS

Reviewer #1 (Remarks to the Author):

The authors of this manuscript describe three strategies for 2P imaging, including a mechanism enabling voluntary head-fixation of rats for seconds to minutes during behavior tasks, a custom chronic window to increase fluorescence detection efficiency, and a transgenic rat line that allows lifespan calcium imaging in hippocampus CA1. While these improvements may be useful for specific applications, I find it challenging to discern the significance of these ideas for broader applications in the field, their innovation compared to current state-of-the-art methods, and the logical relationship between these three aspects—specifically, the necessity to combine them in a single paper. Given these concerns, I would not recommend publication, at least not in journal NC.

Major issues.

I. Voluntary head-fixation part

a. The authors describe a significant advantage of their method, emphasizing reduced stress and ease of use for animals. However, the absence of quantitative data throughout the paper to measure animal behavior during training and experiments makes it challenging to assess whether this method outperforms previous head-fixation techniques, including standard involuntary head-fixation or mechanically clamped voluntary fixation.

We have included additional data to help the reader assess the utility of the magnetic voluntary head-fixation technique. A new supplementary Fig S4 shows the full distribution of all hold periods for all animals, and Fig S6 now includes the distribution of the hold durations for the unconstrained very long hold sessions.

Previous studies of voluntary head-restraint rats and mice have detailed an aversion to mechanical clamping; for instance, Scott 2013, describe an “aversion to the clamp”, and Aoki 2017 describe the steps taken to prevent “a panic reaction, typically accompanied by loud vocalizations and jerky body movements”. Our system eliminates any such aversion by its very nature, since it does not restrain the animals, and they are free to disconnect at any time. The fact that we don’t see such aversive responses is the main evidence of improvement over existing mechanically clamped voluntary fixation.

The other evidence we present that supports our assertion that magnetic fixation is not aversive is the longevity of possible experiments. We have edited Fig3D to better show the cumulative fixation trials for each individual animal. Fig 6E also shows this longevity, showing the imaging sessions for each animal over time.

We believe for these reasons, that our technique is an improvement over existing voluntary head-restraint/fixation systems.

“Standard involuntary head-fixation” is a quite distinct technique with its own pros and cons, animals are typically head-fixed for the whole duration of an experiment, often with VR or some

type of treadmill. We are not proposing that our technique would replace this technique, which has many compelling uses. However, almost all of these experiments are performed in mice; *involuntary head-fixed/restrained* experiments are difficult to perform in rats, as they generally do not tolerate it as well; this is evidenced by the relatively small number of rat involuntary head-fixed studies compared to mouse.

b. The authors claim that previous methods required gradual habituation, but Fig 3B-3C indicates that over 1,000 trials are needed for animals to assume the correct position, which is a substantial training burden. There is no clear evidence of marked improvement in convenience.

Our best assessment of previous work would indicate that our training time is comparable to or an improvement to the previous studies. Comparable to the most similar system in rats, Scott et al., 2013 reports that animals can be trained “in as little as 7 days.” (No data are reported for the average training time). We report an average training time of 12 session, and show an example with 11 sessions over 8 days.

Compared to other, *non-kinematic* systems in mice we demonstrate comparable or improved training times; Murphy et al., 2020 16-25 days, Hao et al 2021 average 7 days, Aoki et al. 2017 not reported, but animals need at least 1 week habituation to apparatus before training even starts to get over their aversion to head-fixation apparatus.

c. The quantitative analysis of variability in holding periods for animals is missing. It's crucial to determine the average holding time within individual animals and across multiple animals, along with its deviation. Inconsistent holding times could be problematic for experiments requiring precise timing, such as closed-loop photostimulation experiments and delay-decision tasks.

We thank the reviewer for this suggestion, and agree that including a description of the variability of holding periods will allow other researchers to evaluate our technique.

We had added a new Figure S4 which shows the distribution of fixation periods for all animal across task performance. We also report the proportion of trials that animals were able to successfully complete the target hold duration on the first attempt. This is an important statistic that that was not included in the original submission, and so we are grateful to the reviewer for raising this point.

d. The repeatability of head-fixation appears convincing but relies heavily on maintaining the stability of the entire system (i.e., x, y, z, and angle positions of the objective and imaging FOV) throughout the experiment period, which spans 1-2 years. Achieving sub-micron level of stability is almost impossible, especially in shared benchtop systems. More importantly, authors did not explain how to use this method for multiple animals when the FOV, imaging depth, or imaging angle are completely different? Did authors mark a specific coordinate for each animals, and adjust the microscope before each experiment? If so, it is hardly to achieve so high repeatability without an online checking, which of course is not compatible with so short head-fixing period (1-2s).

e. Even the reliability of the head-fixation is convincing, but it seems highly depends on the standing-still of the entire system, i.e. the x,y,z and angle position of the objective and imaging

FOV should not be moved during the entire experiment periods (1-2 years). I am afraid this condition is hardly achievable in most of the lab since the benchtop system is usually shared. Even more, how does the author do multiple animals, if the FOV, imaging depth, or even optimized angle are different? One solution may be that a specific coordinate can be assigned to each animal, and before each experiment, the system needs to be adjusted accordingly. But in this way, the registration accuracy is then decided by the repeatability of the imaging system, not the holding mechanism, which will be way more worse. This issue needs to be addressed or at least discussed.

We think that points d and e are duplications of the same question, and so address both below.

We thank the reviewer for raising this issue. There are a number of crucial steps that we took to be able to return to the same field of view that were not explained in the initial submission.

We have included a detailed protocol for how we ensure the registration of the system across the whole experiment. This relies on using a reference calibration slide to zero the system and using the flatness of the slide to check for the angle (which was not typically adjusted). This gets the system into a consistent coordinate frame relative to the kinematic clamp, so that we can return to (roughly) the same field of view. This calibration is performed every time the system is used.

We then use the online field of view registration feature of ScanImage at the start of experiment. We acquire the reference z-stack during an anesthetized session, which allows us to select a field of view. Since the system calibration described above has localized us to almost the correct location the online adjustments we have to make are typically small z position adjustments, and can be easily achieved over ~10 trials with the 2s fixations at the start of each recording session.

Finally, we perform an offline alignment of the fields of view for different imaging sessions using a cross correlation method as described in the methods.

f. The authors mention head-mounted miniscopes for both 1P and 2P systems, which in my opinion could potentially address many of the issues raised. Therefore, the advantages of voluntary head-fixation over miniscope-based freely-moving animals need further demonstration, particularly as newer miniscope systems published in recent years with enhanced scalability and capabilities. Related to e. The biggest advantage of miniscope is that it can continuously recognize for the entire session in any paradigms. But this voluntary head-fixation seems can only be used in a very short fraction of the session, so it can only be used to study very specific questions. The limitations of behavioral paradigms are not discussed in the paper.

We agree with the reviewer that the advances in the development of head mounted microscopes for freely moving behavior are important to mention in comparison to our work. However, we still see advantages that a bench-top imaging system has over the current head-mounted miniscope systems and have expanded our discussion of this in the discussion section to highlight recent developments in two-photon imaging.

One photon mini-scopes cannot provide the optical sectioning of two-photon, and so our system provides an uncontestable advantage in signal-to-noise, anatomical detail and imaging depth. This is especially important for the unambiguous tracking of cells over long periods.

Two-photon head mounted microscope are also an exciting new development, but are still inferior in terms of field of view, numerical aperture (which affects resolution, and signal-to-noise) and collection efficiency.

We have included an additional section in the discussion that highlights the principle limitation of voluntary head-fixation: that neural activity is not recorded all the time. We also include solutions that we and other groups have pursued to address this, for example directional licking.

Voluntary head-fixation has other advantages over freely moving mini-scope imaging, such as being able to perform experiments in home cages with multiple animals (impossible with a cable or tether), and being more suited to high-throughput automated home-cage experiments run without experimenter intervention. The utility of this approach for the neuroscience community is evidenced by other research groups perusing this technology in mice.

II. The conical cannula part

The key result supporting improved collection efficiency in this method is Fig 2D, which appears confusing. It's unclear why the improvement is only observed at the edge (>500um from the center). One possibility is that this improvement results purely from the cropping effect in conventional cylindrical cannulas, which would be naturally resolved by using a conical shape. The contribution of inter-wall reflection remains unclear, requiring more detailed information, such as a comprehensive Zemax/ray tracing analysis of the entire light pathway from the objective to the PMT.

We agree that it is non-intuitive that collection improvement is only observed towards the edge of the cannula, and we thank the reviewer for suggesting approaches to clarify the mechanism at play. We have performed additional bench experiments and new Zemax ray-tracing simulations that together provide a comprehensive and consistent account of the pattern of data we observe.

Firstly, the cropping of the excitation beam could not explain the main effect because the excitation beam is cropped to the same degree for both the conical and cylindrical cannula, as they have the same top diameter. We have revised figure 2 to show the excitation beam for both the cylindrical and conical cannula so that this is clear to the reader.

We also performed additional bench experiments comparing conical and cylindrical cannula with polished, reflective inner walls and with matte black painted walls. These new data are included as a new supplementary Figure, S3. In Fig. S3A using the same conical cannula, comparing polished or painted matte, we see the gain in fluorescence, demonstrating that the effect is a consequence of the reflective inner wall.

Zemax ray tracing

We used non-sequential ray tracing of the cannula, objective and collection system to better understand the pattern of collection improvement with respect to lateral position. This is

presented in a new supplementary Figure S4. With these simulations we were able to recapitulate the pattern of the collection improvement that we measured in the bench experiments: specifically, the maximal improvement towards the edge of the cannula, with little improvement in the center.

These simulation lead to an intuitive explanation for why there is no fluorescence gain in the center of the cannula: reflected rays originating from the center of the cannula exceed the angular acceptance of the objective and collection system. As the fluorescence source moves to the edge, reflected rays decrease in angle and can make it through to the detector.

We have incorporated these findings into the results section and have added points in the discussion section that changing the angle or the shape of the wall could be used to optimize the collection gain for a given field of view location.

Furthermore, conducting simulations in scattering tissue to confirm that this improvement holds within the brain is advisable.

We performed an additional bench experiment (new figure S3,A) with a scattering tissue phantom that was used in previous work to mimic the scattering characteristics of the brain. We saw that the effect was present under the scattering conditions that would be expected in the brain.

III. Lifespan imaging part.

The paper lacks quantification regarding how the same cells are defined, the fraction of cells that can be reliably registered in life-long recordings, and whether this ratio remains stable over different recording intervals. This information is essential for evaluating the reliability and utility of lifespan imaging.

We agree with the reviewer that these are important issues. We have added details in the methods on how the fields of view were registered between session and how we tracked active cells across imaging sessions, for instance for the cells shown in figure 6 and S7.

To help other potential users of this technique assess the potential of this technique, we have included data on all detected cells tracked over the longest time period we report in the supplementary figure S7.

We feel that reporting the fraction of cells tracked dependent on the intervals of recording session would not be useful for readers to assess the methods presented, as there are a number of additional considerations which are not addressed in our current work.

- 1) Our cell detection is activity based, so we only identify ROIs of active cells, that is cells that have significant calcium transients during the recording. This can be seen in figS7F where we can see anatomical correspondence across day in the *no ROI overlap category*; these are cells that are active in only one of the three selected sessions. An accurate assessment of cell tracking would have to consider these partially silent cells, and would require anatomical based correspondence analysis. Consideration of silent cells, and cell

participation over sessions is a research question in its own right (see Perez-Ortega et al 2021 elife).

- 2) We do not have systematic coverage of fields of view with enough different time lags across animals to give a meaningful interpretation of these metrics. Fig 6E shows the full number of sessions over what time period for each animal.

In this manuscript we are showing proof of principle data that these techniques are suited for future, systematic longitudinal studies.

- 1) We show stable fields of view where there is clear correspondence of gross anatomical features and cells identity
- 2) We show sparse expression of GCamp6f that makes tracking of cells much easier since they form unique constellations.
- 3) We show stable expression of the GCamp6f indicated by visible calcium transients and nuclear exclusion of the protein.

We think that the data we present is in line with the current standard of reporting cell tracking in two-photon calcium from other groups. For instance, corresponding fields of view and inspection of ROIs are often presented to support claims of tracking cells over sessions, along with caveats about missing cells.

Driscoll et al 2017 – “Matches [of ROIs across sessions] were then verified by eye”

“However, to prevent such errors [in registration] we visually compared the anatomical images to make sure the signal sources appeared to correspond to the same cell. If a cell could not be confidently identified on a given day, the data were excluded on that day. As a result, our approach resulted in an incomplete map of all cells across all days”

Figure 1 – C fields of view across days,

Pérez-Ortega et al 2021

“we looked at 140 μm depth from pia trying to match the reference image on the x- and y-axis ... Then, we looked for the intersection (in pixels) between ROIs of the neurons from two sessions ($\text{intersection} > 0.5 * \pi * \text{radius}^2$) and evaluate the Euclidean distance between centroids of the ROIs intersected keeping it if the distance $< \text{radius}$.”

Figure 1 – G zoomed in fields of view across days,

Additionally, it is unclear whether all the recordings presented in the paper were conducted under magnetic head-fixation or if these two aspects were separated.

All the data presented are collected under magnetic head-fixation, we have updated the results and figure legends to clarify this.

IV. The logic between these three parts.

The paper suffers from a lack of logical cohesion between the three components: voluntary head-fixation, conical cannula, and lifespan imaging. It appears that these components were assembled without adequate consideration of how they complement each other. Therefore, maybe authors should consider to split them into three separate papers or show a more convincing biological application that cannot be conducted without one of them.

We thank the reviewer for the useful feedback and suggestions, and have attempted to improve the manuscript throughout to link the three components. The techniques presented in the paper, magnetic voluntary head-fixation, conical cannula and transgenic expression of Gcamp6f are all novel steps we took in these experiments; which lead to lifetime imaging of hippocampal neurons. Of course, we can't say that this outcome could not have been achieved without one of these techniques; (these experiments represent a substantial investment of time, so we cannot systematically exclude them) we think that to leave any of these techniques out of the current manuscript would potentially make it more difficult for others to reproduce our work.

We hope that the responses to the reviewer's comments and the other changes and additions to the manuscript have helped to improve our work.

Minor:

In abstract "Existing techniques to record neurons in animals are either unsuitable for 17 longitudinal recording from the same cells or make it difficult for animals to express their full naturalistic behavioral repertoire." the authors should highlight other advantages of their solution instead of claiming that existing techniques make it difficult for animals to express their full naturalistic behavioral repertoire, given the existence of miniscopes.

We have reworded the abstract to emphasize the strengths of the current technique in comparison to existing approaches.

Line 43-45 "but the mechanical solutions implemented to date impose a delay and are a potential point of failure; even one unsuccessful release attempt is enough for the system to become aversive for animals". It's essential to address how their solution avoids these potential pitfalls, especially considering variations in rat size, behavior, and power differences. I can imagine authors still need a lot of trial-and-error and need to adjust the magnetic force rat by rat. During these pilot trials, animals may be stressed.

It is true that a strong magnetic force would prevent a rat from disengaging from the apparatus, and so be potentially, permanently aversive. The magnetic adjustment procedure we describe does not require a lot of trial-and-error, but gradual increase of the magnetic bearing strength from zero to the point where animals can maintain a fixation. We have clarified our procedure for doing this in the methods section. Additionally, we had added data showing the magnetic increase during training in fig.3B. We did not observe aversive responses at any point during the development of this task, the magnetic attractive force we used (which was calibrated with a pull force meter) was never strong enough that the animals could not easily overcome it.

We have included a section in the methods to guide other researchers. Rats can generate pull forces around double their bodyweight. (Giardina et al., 1994), 600g for 300g rats. Our rats weighed more than this (350g-600g), so our 4N maximum pullout force is a reasonable maximum limit for other researcher using rats in this normal weight range.

Line 71-73 “Rats offer the opportunity to study more complex cognitive behaviors compared to mice but previous calcium imaging in rats has relied on the viral expression systems”. This may not be entirely accurate. See these papers from Kerr lab (PNAS 2014 and Nature Methods 2020).

We thank the reviewer for highlighting a previous technique for calcium imaging, We have updated the text to add bolus loading of dye as used in Egger et al., Kerr. “Robustness of Sensory-Evoked Excitation Is Increased by Inhibitory Inputs to Distal Apical Tuft Dendrites.” *PNAS 2015*
The 2020 Nature methods paper (3p head mounted microscope) used a viral expression strategy.

Line 97-98 “evidenced by intermittent failure of the electrical 98 contact mechanism despite sufficient kinematic alignment”. I am not sure if I understand what authors described here. Including a figure and a dedicated method section would enhance clarity.

We have moved this point to the methods where we have gone into detail to better explain the equipment used. We have also included a new figure S8 to illustrate the wearing of the bearing surfaces. This was an infrequent issue in our experiments, and is easily dealt with by replacement of the worn parts. We mention it so that other researchers can recognized if this issue arises and that it can be addressed with replacement of certain parts.

We have also added a discussion point that the system may be further improved by using optical or captative non-contact position sensors rather than the current approach, to sense the head-plate alignment

Line 179 “The gradual introduction of the magnetic force meant that animals were not surprised by it, and we saw no evidence 180 that it was aversive. “ Authors should show supporting data and quantitative analysis are required to validate this assertion..

We have changed the wording here; we do not assert that animals were not surprised, but rather, that we intended it not to be surprising. We have added data to Fig3D showing the introduction of the magnetic force for an example animal. It shows that the animal continues to initiate and complete trails as the magnetic force is introduced.

The lack of aversive response is in contrast of other studies in this field that have reported aversive reactions following or during head-restraint, especially when animals cannot get free. Hao et al., 2021 “mice will start to struggle beyond a certain duration, and if failed to get free, mice will stop engaging in head-fixation subsequently”
Aoki 2017 “[mice show] a panic reaction, typically accompanied by loud vocalizations and jerky body movements”
We did not observe any such responses at any point in this study.

Fig.2B. This figure appears to be an illustrative cartoon, and it does not seem to represent an actual ray-tracing simulation. Given that the authors mention the use of an air objective, the angle of ray traces shown here seems already beyond the total-internal reflection criteria and should not be able to exceed the glass window. To ensure clarity and accuracy, it is recommended that the authors provide a genuine ray-tracing simulation that reflects the conditions mentioned in the manuscript. This would help readers better understand the optics involved in their methodology.

We have updated Fig. 2 to better show the reader the relevant optics. Ray tracing in Zemax was used to generate the two panels Ai and Aii. The total internal reflection condition of the emission fluorescence beyond the critical angle is evident.

Finally, we want to thank the reviewer for the carefully thought-out comments that are clearly designed to improve the science and communication of our work.

Reviewer #2 (Remarks to the Author):

The manuscript “Magnetic voluntary head-fixation in transgenic rats enables lifetime imaging of hippocampal neurons” by P.D. Rich and the coauthors is well written and clearly organized. In this manuscript, Rich et al. introduce a novel approach called the "magnetic voluntary head-fixation system" for conducting voluntary head-fixed, long-term two-photon imaging in transgenic rats. The paper presents several significant technical advancements compared to existing methodologies: 1) Kinematic Mount with Kelvin Coupling: The authors employ a kinematic mount based on the Kelvin coupling, providing micron-scale registration accuracy between insertions. 2) Magnetic Head Fixation: Instead of using conventional mechanical clamps, the authors utilize a magnetic fixation approach. This magnetic system offers several advantages, including ease of use for animals, reduced aversiveness, and the elimination of potential failures associated with moving parts. 3) depth Conical Cannula: To enhance emission signal collection, the authors introduce a reflective conical cannula. This modification significantly improves signal collection efficiency compared to traditional cylindrical cannulas. The authors apply their magnetic head-fixation system to Thy1-GCaMP6f transgenic rats and demonstrate the system's adaptability and long-term stability over the animals' lifetimes. Two-photon imaging is conducted during fixation while the rats perform a place-odor association task, leading to the identification of odor-responsive cells. This work holds substantial promise for the field of neuroscience, making it well-suited for publication in Nature Communications. However, there are several important issues that need to be addressed to enhance the manuscript's quality.

1) Figure 2B: It is advisable to label the schematics in Figure 2B as "Nonreflective Cylindrical" and "Reflective Conical" to clarify the critical role of the reflective inner surface in enhancing signal collection.

Additionally, it would be valuable to assess how much the conical shape improves signal collection compared to both cylindrical and conical shapes with reflective inner surfaces. Performing non-sequential optical simulations and comparing them with experimental data could provide valuable insights.

We agree that this is an important issue. We have revised Fig. 2, including the suggested labeling. We performed new bench experiments and ray tracing simulations to clarify the contribution of the shape of the cannula and the reflectivity of the wall.

We performed additional analysis, using non-sequential ray tracing of the cannula, objective and collection system in Zemax to better understand the lateral pattern of collection improvement. This is presented in a new supplementary Fig S4. The simulations recapitulated the effect we saw in the bench experiments, and demonstrates that the gain in fluorescence collection is a consequence of the reflective surfaces.

We performed additional bench experiments comparing the same conical and cylindrical cannulas in a mirror polish state or painted matte black. These experiments are presented in a new supplementary Fig. S3. Comparing the same, either polished or matte black painted conical cannula confirmed that the reflective inner wall underlies the gain in fluorescence observed.

We also saw a very small increase in fluorescence in the cylindrical polished vs matte state, at the very edge of the cannula (which was actually predicted by the ray-tracing). The increase is lower than observed in simulation since the fluorescence is decreasing due to clipping of the excitation beam. This observation also provides an independent validation of our ray-tracing simulation.

2) Figure 2D: Provide information on the internal diameter of the cylindrical cannula and how it compares with the conical cannula in this test. Explain why the red plot ends before reaching 2 mm while the cylindrical one reaches almost 2.5 mm. Address whether the excitation beam cone is obstructed by either of the two cannulas during this test. Also, ensure consistent terminology by using "Nonreflective Cylindrical" and "Reflective Conical" instead of "Cylindrical" and "Conical."

We have added the dimensions of the cylindrical cannula used in the tests to the methods. The inner diameter of the cylindrical cannula is within 0.1mm of the conical cannula, and this has been clarified in the methods section. We have also added a supplementary Fig S2 which is the fully dimensioned technical drawing of the conical cannula geometry.

We have also updated Fig.2 with accurate ray-traced diagrams of the conical and cylindrical cannulas. We hope that this updated figure addresses the reviewer's points.

-It shows that the two cannulas have the almost the same top diameter.

-It shows the excitation beam and how it would be equivalently obstructed in both conditions.

-The labels have been changed to indicate reflective vs non-reflective conditions.

-It illustrates why the conical cannula collected Fluorescence plot only extends to ~2mm while the cylindrical cannula goes to 2.5mm; this is because the bottom diameter of the conical cannula is smaller. The hatched invalid region is described in the caption.

The obstruction of the excitation beam as a function of lateral position can be seen from the ray-tracing data (Fig S4); because of the principle of reversibility, the total luminous flux in the absorb condition is equivalent to the obstruction.

3) Figure 2D's Caption: Clarify whether the signal is measured at 150 μm below the cover glass as stated in the caption or if the average signal from different depths is quantified, as mentioned in the Methods section. Discuss whether the improvement in collection efficiency is sensitive to the imaging depth and, if so, identify the optimal depth.

We did indeed collect the data for different imaging depths. The data shown in Fig. 2 is for one depth, 150 μm , which is approximately the depth that the pyramidal cell layer is visible. We have included an additional figure (Fig S3 C,D) which shows the full data for various depths up to 450 μm deep. From this figure, we can see that the profile of fluorescence gain changes with depth, but has roughly the same shape. The depth that shows the highest (relative) gain is the shallowest depth.

We also added a point in the discussion, that shape of the cannula wall could be optimized to maximize collection efficiency for a given location and depth. This is an area that we plan to pursue in the future.

4) Figure 3B: Address the observed transition between stages 1 and 2, which appears abrupt and suggests a possible nonvoluntary factor such as applied force. Provide data on the force applied during the training stages and its duration, accompanied by the original Figure 3B for clarity.

The abrupt transition at the start of stage 2

corresponds to the animal learning to make the “*contact*” required action (for stage 2, that is contacting the rear bearing) which leads to an immediate double reward. The abrupt jump shows that the animal has learned that it does not have to maintain the full hold time for the hold action, but can just make a contact. This is facilitated by the weak magnetic force at the rear bearing, but, as in all experiments, the animal can easily break contact (actually has to be trained to maintain the hold). We have updated the figure and the legend to better explain this.

We have also updated the figure to include the magnetic force levels that were set for each bearing and the nose poke angle for each trial in the training.

5) Headplate Weight: Specify the weight of the head plate used in the experiments.

We had added this detail to the methods section

6) Magnetic Force and Contamination: Discuss the potential issue of small objects, such as bedding debris, contaminating the bearing surfaces and possibly interfering with the magnetic connection. Clarify whether such occurrences happened during the experiments and their frequency.

This is an important point and we thank the reviewer for asking about it, as it will make future work easier for others. We had added the details of how we cleaned the bearing surfaces before

each experiment to the methods section. This cleaning protocol ensured that there was a clean bearing surface that enabled reliable electrical and magnetic functionality.

7) Baseplate and Cannula Alignment: Explain the methods employed to ensure concentricity and parallel alignment between the baseplate and the cannula during the craniotomy process.

For the concentricity between baseplate and cannula, alignment was not need to be especially accurately ensured. We centered by eye the aperture of the baseplate and the implanted cannula during surgery. Since the microscope can move in x and y relative to the kinematic mount, this was sufficient to ensure that we could image anywhere on the cannula bottom,

Parallel alignment: we added a section of the methods to describe how this was achieved during surgery. Both the canula and baseplate were implanted using fixtures that ensured they were fixed parallel to the stereo-tax, and so to each other.

We have updated the methods to describe this.

Reviewer #3 (Remarks to the Author):

This manuscript describes a system for automatic head fixation of rats that employs permanent magnets. the system is an adaptation of a previous publication from the group on a kinematic mount for headfixing mice that was published in the journal of neuroscience methods by Scott. This paper by the originators of the automatic headfixation technique in rats significantly extends the mouse magnet work because now they show the system is viable for collecting long term gcamp fluorescence data. Light collection is improved using a new canula.

This is exceptional work and represents a direction the field should be going-automated experiments controlled by the animal. In general is put together well and contain strong data showing feasibility. However, I feel as a method's paper it falls short because there is not enough information here for one to replicate the findings. There are some images of the system, but full mechanical drawings, software, and procedures are lacking they should be made available and linked to the submission directly.

We thank the reviewer for the positive feedback on our experiments and approach.

We have taken to heart the recommendation to provide more information about the techniques, as we agree that this is a useful direction for the field. We hope that have now provided sufficient information that other users would be able to recreate out approach, and we are very grateful that we have a chance to improve on the initial submission.

Throughout the manuscript we have tried to provide additional details on procedures, part numbers and suppliers.

We have created a public and open source licensed GitHub repository that contains the information needed for other researchers to recreate or event adapt or redesign the system.

<https://github.com/dylan2106/Magnetic-Voluntary-Head-fixation>

The address for the GitHub repository is listed at the end of the methods section. We include the full CAD files for the magnetic voluntary head-fixation system, which will allow other researchers both to manufacture their own systems as we have presented, as well as adapt or

change the design for their own use. In the top level readme we provide notes on the construction and assembly of certain particular components.

Here is a link for an online viewer of the main assembly of the CAD <https://autode.sk/3uTGcX5> (This link is for the convenience of the reviewer to view the CAD files without installing any software. Unfortunately, Autodesk will only host this online view at that link for only 30 days. All the files are downloadable permanently from GitHub though.)

In the Github repository, we also included the custom firmware, software and PCB design files we used for our system. We have tried to organize the repository to be modular so that users may take and adapt whatever aspect of the system they need. For instance, the hardware for the kinematic system may be integrated into other experimental control systems since it minimally requires only some form of current sensing through the bearings and detection of an infrared beam break in the nose poke.

It should be clearly stated that the magnets employed are permanent and not electromagnets, minor.

We had specified the magnets as being permanent magnets where relevant.

Registration and motion correction is brought up, but it is not clear how registration is done for repeated 2-photon imaging, and if it would be possible to automatically register animals that are imaged on different days? the paper implies that this is done, but there should be details given as this is key, also please share software for registration and control (is this via scanimage)?

We have added a section in the methods detailing the field of view registration across days. The main technique was to use the built-in registration tools in scanimage to return to a target field of view. The scanimage tools require a reference z-volume to be taken, which we performed in an anesthetized recording session. For small offline correction shifts we just used a simple cross correlation approach detailed in the methods.

Figure 3 is interesting, showing hold time. Data in panel B and total trials over up to 18 months, which is absolutely unprecedented. However, there is no mention of the sample size for these measurements (animal number for B,C, and D please). Is this a single rat or is it group data. We have added clarification that the data for C & D are for all animals (B was already indicated for a single animal). We have also updated the graphic in D to show separate animals more clearly. Additionally, all the sessions where we recorded calcium data (which were a subset of behavioral sessions) are shown for all animals in Fig. 6E.

the trials are also relatively short in duration only 2 seconds. I think we would like to see a distribution of trial length across the different animals, what duration do rats prefer? It would also be good to separate male and female data.

We have included the distributions of the fixation durations for all animals as a new supplementary figure S4. This shows that actual distributions of the fixation durations across many months of task performance. We have noted the sex of individual animals in the figure, indicating that both male and female animals can achieve successful fixations.

There is a section on long fixation. But this doesn't have any sample sizes, so we don't know if this is one rat or a group, this should be clearly indicated and all data shown.

We have clarified this in the results and the figure legends.

Two animals were trained on 4s duration holds, and one animal was trained on unconstrained holds.

Fig S2 is nice for long holds but this is just one rat and it does not indicate a distribution of fixation times, based on the data shown 37 sec was the longest hold time. Please show more stats.

We have updated figure S2 S6 to show the distribution for all fixation for this animal on indefinite long fixation sessions. The longest fixation was 70 sec. There was no real extra incentive for the animal to remain fixated for these very long periods, as the reward was delivered at random intervals while the animal-maintained fixation.

Very short hold times may limit the utility of the method as movement related activity will likely contaminate recordings. please discuss and present all long hold data. Ideally, one would expect to see a table with all of the animals and something like the duration of fixation, numbers of trial, etc like the data in Fig. 3 but group stats.

We present all the unconstrained long hold data in the updated figure S2 S6.

In the new figure S5 we show individual trials over a number of sessions of one of the animals trained to go for 2.5s > 4s hold duration. We also show the full distributions of fixations across all animals for the whole experiment. Here we show that animals continue to make the fixation following the required hold time. We report the proportion and show the distribution of all the first attempt fixations, as well as the completed fixations (which includes additional attempts the animal make). On average, animals make the make the required fixation duration on the first attempt in 76% of trials.

The method section needs some addition to work to better. Describe novel elements and critical components. Many items are mentioned by supplier only, no catalog number or model number, the J Neurosci Meth article is better done in this regard. This includes the permanent magnet cat # and micrometers as well as critical ball bearings.

We included the catalog numbers for the permanent magnets, micrometers, bearing balls and other key parts. We have included additional details throughout the methods to better describe the system and make it more straight-forward to reproduce. We also included a bill of materials for critical components in the linked GitHub repository.

The possibility of using similar mounts on mice is also mentioned but I feel given the work that has been done by some of the co authors in mice that the issue of mice could be addressed more thoroughly, how would the rat system shown be modified and what are the holding forces etc? Can mice be trained for long holds?

The Kim et al. 2023 study only presents a kinematic mount, it has no voluntary fixation element. The design of the voluntary aspect of our system is a large design consideration compared to a kinematic mount alone. We have not performed any experiments with mouse voluntary head-

fixation (The co-author mentioned, B.B.S., contributed to the current study with transgenic animal screening, the Kim et al., study was performed separately in their own separate research group). We would expect that mice would be able to use a scaled down system given the prior research with mechanical clamps where mice are able to perform long holds (Murphy et al 2020, Aoki et al. 2017, Hao et al., 2021). This is an important avenue for future research.

Provide a bit of information about the air objective used X,Y, Z resolution as they have used it for 2P in the current rig.

We had included the measured resolution of the system in the methods; the FWHM of the PSF, which was 1.2 and 9.1 μ m.

Minor

“Figure 6 - Lifetime imaging of the same cells” lifetime imaging could mean fluorescence lifetime so this might confuse some?

We changed this to “Lifetime, longitudinal imaging...”

REVIEWER COMMENTS

Reviewer #1 (Remarks to the Author):

The authors have addressed most of my previous questions or concerns. The current version is much improved with much more details and better logical links of three features. I particularly liked the below augments that the authors made in their rebuttal letter that "...involuntary head-fixed/restrained experiments are difficult to perform in rats, as they generally do not tolerate it as well; this is evidenced by the relatively small number of rat involuntary head-fixed studies compared to mouse.". I think many people may not be aware of this issue so I highly suggested the authors to put this argument in the discussion or somewhere else. Beside that, I have no more comments.

Reviewer #2 (Remarks to the Author):

The authors have addressed all my questions.

Reviewer #3 (Remarks to the Author):

The authors have addressed most of my concerns this is an important study and a step forward in terms of how experiments are conducted.

Looking at the distribution of hold times/trial lengths one is struck by how short they are. The abstract mentions long-term experiments but this term is potentially confusing, as there are months or experiments but only seconds of each trial.

"This system is failsafe, easy for animals to use and reliable enough to allow long-term experiments to be routinely performed."

I think the authors should give some maybe median times for continuous longitudinal imaging (many months so impressive) and also the median trial length for a single head-fixation (a few seconds) in the abstract. I think if this is given it would present it quite transparently. I understand that rats can be trained to fix for longer and have more complex trials but it seems they like short bouts; the training for longer fixation can also be emphasized in the paper.

The issue of movement-related brain activity when repeatedly fixing could be acknowledged as a confound of short-duration trials.

Reviewer #3 (Remarks on code availability):

Github repo has schematics and code

REVIEWER COMMENTS

Reviewer #1 (Remarks to the Author):

The authors have addressed most of my previous questions or concerns. The current version is much improved with much more details and better logical links of three features. I particularly liked the below augments that the authors made in their rebuttal letter that "...involuntary head-fixed/restrained experiments are difficult to perform in rats, as they generally do not tolerate it as well; this is evidenced by the relatively small number of rat involuntary head-fixed studies compared to mouse.". I think many people may not be aware of this issue so I highly suggested the authors to put this argument in the discussion or somewhere else. Beside that, I have no more comments.

We agree with the reviewer that this is an important point and have included it in the discussion section.

Reviewer #2 (Remarks to the Author):

The authors have addressed all my questions.

Reviewer #3 (Remarks to the Author):

The authors have addressed most of my concerns this is an important study and a step forward in terms of how experiments are conducted.

Looking at the distribution of hold times/trial lengths one is struck by how short they are. The abstract mentions long-term experiments but this term is potentially confusing, as there are months or experiments but only seconds of each trial.

"This system is failsafe, easy for animals to use and reliable enough to allow long-term experiments to be routinely performed."

I think the authors should give some maybe median times for continuous longitudinal imaging (many months so impressive) and also the median trial length for a single head-fixation (a few seconds) in the abstract. I think if this is given it would present it quite transparently. I understand that rats can be trained to fix for longer and have more complex trials but it seems they like short bouts; the training for longer fixation can also be emphasized in the paper.

We have updated the abstract to address the reviewer's concerns, and clarify that we have short duration trials, over long periods of the animals' lives. We have included the range of values that were *required* for animals to fixate (not including the additional time the animals remained fixed to consume the reward, the full distributions are in figure S5). We have also included the median length of time that we were able to longitudinally record from the hippocampus as per the reviewer's suggestion.

The issue of movement-related brain activity when repeatedly fixing could be acknowledged as a confound of short-duration trials.

We have added this point in the discussion.

Reviewer #3 (Remarks on code availability):

Github repo has schematics and code

REVIEWERS' COMMENTS

Reviewer #3 (Remarks to the Author):

The authors have addressed all my concerns and I recommend publication

Reviewer #3 (Remarks on code availability):

well documented